# IMPMCT: a dataset of Integrated Multi-source Polar Mesoscale Cyclone Tracks in the Nordic Seas

Runzhuo Fang<sup>1,2</sup>, Jinfeng Ding<sup>1,2</sup>, Wenjuan Gao<sup>1,2</sup>, Xi Liang<sup>3</sup>, Zhuoqi Chen<sup>4</sup>, Chuanfeng Zhao<sup>5</sup>, Haijin Dai<sup>1,2</sup>, Lei Liu<sup>1,2</sup>

<sup>1</sup>College of Meteorology and Oceanography, National University of Defense Technology, Changsha, China

<sup>2</sup>Key Laboratory of High Impact Weather(special), China Meteorological Administration

<sup>3</sup>Key Laboratory of Marine Hazards Forecasting, National Marine Environmental Forecasting Center, Ministry of Natural Resources, Beijing, China

<sup>4</sup>School of Geospatial Engineering and Science, Southern Marine Science and Engineering Guangdong Laboratory (Zhuhai), Sun Yat-sen University, Zhuhai, China

<sup>5</sup>Department of Atmospheric and Oceanic Sciences, School of Physics, and China Meteorological Administration Tornado Key Laboratory, Peking University, Beijing, China

15 Correspondence to: Jinfeng Ding (dingjinfeng@nudt.edu.cn)

Abstract. Polar Mesoscale Cyclones (PMCs), particularly their intense subset known as Polar Lows (PLs), characterized by short lifespans of 3-36 hours and horizontal scales below 1,000 km, pose significant hazards to polar maritime activities due to extreme winds exceeding 15 m s<sup>-1</sup> and wave heights surpassing 11 meters. These intense weather systems play a critical role in modulating sea-ice dynamics and ocean-atmosphere heat exchange. However, current understanding remains constrained by sparse observational records and an overreliance on single data sources (e.g., remote sensing or reanalysis). To address these gaps, this study presents the Integrated Multi-source Polar Mesoscale Cyclone Tracks (IMPMCT) dataset, a comprehensive 24-year (2001-2024) record of wintertime (November-April) PMCs for the Nordic Seas. The IMPMCT dataset was created by combining vortex-tracking algorithms applied to ERA5 reanalysis data with a deep learning-based method for detecting cyclonic cloud features in Advanced Very High-Resolution Radiometer (AVHRR) infrared imagery. It also incorporates nearsurface wind data from Advanced Scatterometer (ASCAT) and Quick Scatterometer (QuikSCAT) measurements. The dataset contains 1,110 vortex tracks, 16,001 cyclonic cloud features including length, width, position and morphological characteristics (spiral/comma shape), and 4,472 wind speed records (wind vector imagery and cyclone maximum winds). Corresponding ERA5-derived hourly vortex tracks are also provided, including 850 hPa vorticity and proximate sea-level pressure minima. Validation demonstrates statistical agreement with existing PLs track datasets while providing more complete cyclone life cycle trajectories, more intuitive cloud imagery visualization, and a richer set of parameters compared to previous datasets. As the most comprehensive PMCs archive for the Nordic Seas, the IMPMCT dataset provides fundamental data for advancing our understanding of the genesis and intensification mechanisms, enables the development of enhanced monitoring and early warning systems, supports the validation and refinement of polar numerical weather prediction models, and facilitates improved risk assessment and safety protocols for maritime operations. The dataset is available at https://doi.org/10.5281/zenodo.17142448 (Fang et al., 2025).







#### 1 Introduction








Polar Mesoscale Cyclones (PMCs) are mesoscale cyclonic weather systems that frequently occur over open waters or sea-ice edges in regions poleward of the main polar front zones (Condron et al., 2006; Rasmussen and Turner, 2003). They are often identified on satellite imagery by comma-shaped or spiral cloud patterns. PMCs occur in all seasons but are most active in winter, with a lifespan of approximately one day and horizontal scales of less than 1,000 km (Harold et al., 1999). The most intense subset of these cyclonic systems, termed Polar Lows (PLs), are major hazardous weather phenomena in polar regions, characterized by average maximum wind speeds exceeding 15 m s<sup>-1</sup> and extreme values surpassing 30 m s<sup>-1</sup>. They can generate significant wave heights over 11 meters (Rojo et al., 2019), posing severe threats to human activities and maritime safety in high-latitude regions (Harrold and Browning, 1969; Orimolade et al., 2016). Additionally, PLs induce rapid sea-ice changes and intensify ocean-deep convection through dynamic and thermodynamic effects, producing complex regional climatic impacts (Clancy et al., 2022; Condron and Renfrew, 2013; Parkinson and Comiso, 2013). The Nordic Seas (encompassing the Greenland, Norwegian, and Barents Seas) form a critical oceanic gateway between the Arctic and Atlantic Oceans. This region is a primary convergence zone for Arctic and Atlantic water masses and plays a key role in global ocean circulation and climate (Smedsrud et al., 2022). The complex meteorological and oceanographic conditions in this area make it the most frequent PLs occurrence region (Stoll, 2022). Consequently, research on mesoscale cyclones in the Nordic Seas is critical for improving Arctic maritime safety and understanding regional climate change impacts.

Cyclonic cloud morphology and surface wind fields derived from remote sensing data serve as the primary criteria for distinguishing and categorizing PMCs and PLs (Rasmussen and Turner, 2003). The former can be manually identified through visible or infrared imageries from passive radiometers (e.g., Fig. 1), while the latter can be estimated using scatterometer or microwave data. While PLs exhibit higher destructive potential and detection feasibility compared to broader PMCs, current dataset development efforts have predominantly targeted PLs, leaving PMCs relatively underrepresented in existing observational records. Blechschmidt et al. (2008) combined Advanced Very High-Resolution Radiometer (AVHRR) infrared imagery (Kalluri et al., 2021) with wind speed data derived from the Hamburg Ocean Atmosphere Parameters and Fluxes from Satellite Data (HOAPS, Andersson et al., 2010) to manually identify 90 PLs occurring in the Nordic Seas between 2004 and 2005. Noer et al. (2011) utilized AVHRR infrared imagery, Advanced Scatterometer (ASCAT), and Quick Scatterometer (QUIKSCAT) wind data to detect 121 PLs in the Nordic Seas over a decade (2000-2009). Smirnova et al. (2015) identified 637 PLs between 1995 and 2009 using Special Sensor Microwave/Imager (SSM/I) data for atmospheric total water vapor (TWV) content fields, near-surface wind speed fields, and AVHRR infrared imagery. Golubkin et al. (2021) employed Moderate Resolution Imaging Spectroradiometer (MODIS) and ASCAT data to identify PLs over the North Atlantic, compiling a catalog of 131 PLs between 2015 and 2017. In all PL lists derived from remote sensing data, the Rojo list (Rojo et al., 2015, 2019) is currently the longest temporally spanning remote sensing-derived PLs track dataset, providing tracks of 420 PLs occurring in the Nordic Seas from 1999 to 2019. It includes basic information such as cyclone location, size, type, development stage, and maximum 10 m wind speed. The manually tracked datasets described above have provided valuable PLs information, contributing to ongoing research efforts. However, the unique high-latitude geography of polar regions creates significant observational challenges. Polar-orbiting satellites typically observe these regions at intervals ranging from tens of minutes to several hours, resulting in temporal gaps that make it difficult for manual tracking datasets to capture complete cyclone life cycles. Additionally, some PLs forming near sea-ice edges may exhibit distinct cyclonic cloud features exclusively during their transition over moisture-rich open waters (Bromwich, 1991), implying that remote sensing datasets could potentially miss capturing the initial developmental stages of such PLs. Consequently, while the Rojo list provides developmental pattern annotations for individual PLs, the objectivity and quantitative reliability of these annotations remain constrained by the inherent limitations of remote sensing in achieving comprehensive characterization of PL evolution throughout their complete lifecycle. Furthermore, the occurrence of polar night, coupled with low contrast between sea-ice/snow surfaces and overlying clouds, further limits the detection capabilities of remote sensing (particularly visible-band remote sensing) methods for PLs.

Figure 1 Two AVHRR satellite images. (a) A PMC in Barents Sea. (b) A PL in Norwegian Sea. The yellow stars mark the centers of these two cyclones.

With the improved resolution of reanalysis datasets, their ability to characterize PLs has progressively advanced (Laffineur et al., 2014; Smirnova and Golubkin, 2017), making them an increasingly critical data source for constructing PLs track datasets. Researchers have employed various combinations of identification criteria to detect PLs. For instance, Zappa et al. (2014) utilized the difference between 500 hPa temperature and near-surface temperature to represent cold air outbreak characteristics during PLs formation, while utilizing maximum near-surface wind speed to indicate PLs intensity, and 850 hPa relative vorticity to capture their cyclonic properties. Subsequent studies adopted or adapted these criteria (Stoll et al., 2018; Terpstra et al., 2016; Yanase et al., 2016). Building on the fifth-generation European Centre for Medium-Range Weather Forecasts Reanalysis (ERA5, Hersbach et al., 2020), Stoll (2022) established a four-criteria linear-based combination defining PLs as intense mesoscale cyclones forming within polar oceanic air masses northward of the polar front. This approach successfully reproduced 60-80 % of PLs from five manual PL lists, validating ERA5's robust capability

in PLs representation. However, ERA5 significantly underestimates near-surface wind speeds within PL-affected regions (Gurvich et al., 2022; Haakenstad et al., 2021), attributed in part to insufficient representation of transient wind variability, surface divergence, and unresolved mesoscale features (Belmonte Rivas and Stoffelen, 2019). This limits its ability to objectively capture PLs' high-wind characteristics, thereby introducing notable limitations.

In summary, remote sensing and reanalysis datasets provide complementary perspectives on PLs' characteristics, with the former capturing cloud morphology and the latter resolving meteorological field distributions, highlighting their respective advantages. This complementary nature motivates the integration of both data types to construct more comprehensive PL tracking datasets—a key objective of this study. Furthermore, existing datasets primarily focus on PLs, while weaker PMCs that share similar cyclonic cloud features and environmental conditions lack comprehensive publicly available track datasets. This disparity likely stems from the fact that PMCs generally have smaller average intensities, shorter lifespans, and smaller scales compared to PLs, making them more difficult to detect. Although some researchers have proposed PMC track datasets using either remote sensing (Verezemskaya et al., 2017) or reanalysis data (Michel et al., 2018; Pezza et al., 2016; Watanabe et al., 2016), these approaches face significant limitations. Remote sensing-based datasets often have inadequate temporal coverage or lack critical near-surface wind speed records (Condron et al., 2006), while reanalysis-based datasets encounter challenges in developing effective identification criteria without remote sensing validation. As a result, no universally accepted PMC identification standards currently exist (Michel et al., 2018). Notably, while PLs have been well-documented in relation to large-scale circulation patterns such as the North Atlantic Oscillation (Claud et al., 2007) and Scandinavian blocking (Mallet et al., 2013), the climatic impacts of PMCs remain insufficiently investigated (Michel et al., 2018). These knowledge gaps highlight the critical need to establish a more comprehensive tracking dataset capable of capturing PMCs throughout their lifecycle. Such a dataset would enable the complete characterization of these weaker polar mesoscale systems, representing another key motivation for this study.

Based on the above analysis, this study aims to comprehensively integrate the advantages of reanalysis datasets in characterizing the dynamical and thermodynamic structures of polar mesoscale weather systems and remote sensing data in capturing cloud morphology to establish a long-term PMCs (hereafter, "PMCs" when used alone include "PLs") track dataset in the Nordic Seas encompassing the extended winter seasons (November-April) between 2001 and 2024. This dataset will contain the tracks of the PMCs in reanalysis fields and remote sensing imagery, as well as multi-dimensional attributes such as intensity, cloud morphology, and near-surface wind features. The objective is to provide a long-term, multi-attribute catalog of PMCs, offering reliable data support for atmospheric and oceanic research in the Nordic Seas.

#### 2 Data









## 2.1 AVHRR data

The Advanced Very High-Resolution Radiometer (AVHRR) (Kalluri et al., 2021) is mounted on NOAA series meteorological satellites and MetOp series satellites. Since its launch with the TIROS-N

satellite in 1979, the sensor has continuously performed multiple daily observations of the Earth's surface. It measures reflected and emitted radiation from the Earth and its atmosphere, providing detailed information about surface characteristics, clouds, and atmospheric properties. AVHRR is an across-track scanning system with five spectral bands as shown in Table 1. It has a nadir spatial resolution of approximately 1.1 kilometers and a  $\pm 55.4^{\circ}$  scan angle on the satellite, covering a ground swath width of 2,800 km. However, the effective resolution depends on the scan angle, with optimal image quality provided within the  $\pm 15^{\circ}$  range.

In this study, infrared imagery used to observe cyclonic cloud features is derived from two Level 1B data products of the AVHRR (Kalluri et al., 2021): the GAC (Global Area Coverage) and LAC (Local Area Coverage) forth-band data. The GAC product provides down-sampled imagery (approximately 4 km resolution) after onboard processing, selecting every third scan line and averaging every fifth adjacent sample along the scan line. This resampling aims to ensure continuous global coverage. In contrast, the LAC product records AVHRR data at its native resolution (1.1 km) without resampling over specific orbital regions (primarily Europe and Africa), offering higher spatial resolution. All AVHRR data utilized herein are obtained from NOAA's Comprehensive Large Array-data Stewardship System (https://www.aev.class.noaa.gov/ (accessed on 18 July 2024)).

Table 1 AVHRR radiometer channel information.

| Table 1717 Hitte Tadionicter Chamier mior mation. |                        |            |                         |                                                         |  |  |
|---------------------------------------------------|------------------------|------------|-------------------------|---------------------------------------------------------|--|--|
|                                                   | Channel Wavelength(μm) |            | Satellite               | Application                                             |  |  |
|                                                   | 1                      | 0.58-0.68  | ALL satellites          | Surface albedo estimation                               |  |  |
|                                                   | 2                      | 0.725-1.00 | ALL satellites          | Water body delineation                                  |  |  |
|                                                   | 3A                     | 1.58-1.64  | NOAA15-<br>19/MetOP A-C | Snow and ice cover identification                       |  |  |
|                                                   | 3B                     | 3.55-3.93  | NOAA8-<br>19/MetOP A-C  | low-level clouds identification and surface temperature |  |  |
|                                                   | 4                      | 10.3-11.30 | ALL satellites          | Cloud-top temperature and surface temperature           |  |  |
|                                                   | 5                      | 11.50-12.5 | NOAA8-<br>19/MetOP A-C  | Cloud-top temperature and surface temperature           |  |  |

#### 2.2 ERA5 data






ERA5 is the fifth-generation global reanalysis dataset produced by the European Centre for Medium-Range Weather Forecasts (ECMWF), designed to provide high-quality, consistent estimates of atmospheric, land, and ocean climate variables from 1950 to the present. It replaces the previous ERA-Interim dataset (Dee et al., 2011) and is currently one of the most widely used reanalysis products. ERA5 offers hourly data with a horizontal spectral truncation of T639, corresponding to a global grid of approximately 31 km. The atmosphere is resolved vertically using 137 levels extending from the surface to 80 km in height (Han and Ullrich, 2025).

ERA5 reanalysis dataset demonstrates robust performance in representing meteorological fields over the Nordic Seas, such as sea level pressure, air temperature, and humidity (Graham et al., 2019; Moreno-Ibáñez et al., 2023; Yao et al., 2021). Most notably, its effective characterization of cold air outbreaks has been proven to correlate closely with the timing and location of PLs (Meyer et al., 2021).

However, beyond the previously mentioned underestimation of near-surface strong winds in Sect. 1, Wang et al. (2019) found ERA5 data exhibits a warm bias over Arctic sea ice during winter and spring, which makes it difficult to accurately simulate the frequently occurring strongly stable boundary layers prevalent in winter and early spring. Consequently, the intensity of PMCs near the sea ice edge might be overestimated. Nevertheless, more accurate total precipitation and snowfall data in ERA5 (Wang et al., 2019) significantly benefits the representation of enhanced latent heat release mechanisms associated with PLs (Moreno-Ibáñez et al., 2021).

In this study, we utilize ERA5 reanalysis data spanning 2001-2024 during the extended winter period (November-April), on a spatial grid of  $0.25^{\circ} \times 0.25^{\circ}$ , covering the domain  $50^{\circ}$  N-85° N in latitude and  $40^{\circ}$  W-80° E in longitude. This dataset is employed to track vortices and compute their evolutionary characteristics such as intensity and size.

#### 2.3 QuikSCAT/ASCAT data









This study further leverages QuikSCAT and ASCAT data to examine near-surface wind field properties within the cyclone core and its surrounding ambient conditions. QuikSCAT, a NASA-developed Earth-observing satellite, employs a Ku-band SeaWinds microwave scatterometer to provide global measurements of ocean surface wind vectors. Similarly, ASCAT features a C-band microwave scatterometer aboard EUMETSAT-operated MetOp polar-orbiting meteorological satellites. These advanced instruments are specifically engineered to deliver accurate (e.g., ASCAT-A zonal/meridional wind component error standard deviations of ~0.37/0.51 m s<sup>-1</sup> and ASCAT-B of ~0.39/0.44 m s<sup>-1</sup>, Vogelzang and Stoffelen 2022), high-resolution, continuous wind vector measurements under all weather conditions, offering comprehensive global coverage of near-surface wind patterns. The full potential of these measurements extends to their spatial derivatives, specifically vorticity and divergence, which are closely associated with deep moist convection and cyclonic activity (King et al., 2022).

We utilize Level 2 near-surface wind vector retrieval products from both instruments to analyze wind field characteristics during cyclone development, with both datasets featuring a 12.5 km resolution. For QuikSCAT, a slice-based compositing technique integrates high-resolution measurements derived from Level 1B data into 12.5 km wind vector cells. In contrast, ASCAT employs a spatial box filter to minimize land contamination of microwave signals and enhance retrieval accuracy in coastal regions. Both datasets are sourced from NASA's Physical Oceanography DAAC (podaac.jpl.nasa.gov/ (accessed on 28 November 2024)). For the two products, QuikSCAT is available from 1999 to 2009, whereas ASCAT start providing since 2010. To ensure comprehensive temporal coverage across the track dataset, the two products are utilized in their respective operational periods to ensure comprehensive temporal coverage.

## 3 Methodology

To establish a more comprehensive cyclone track dataset in the Nordic Seas, we first utilize ERA5 reanalysis data which exploits the evolving global observing system to obtain all vortex tracks. In this process, a lower vorticity maxima criterion is applied to extract vorticity perturbations within the

reanalysis data. Subsequently, vortex tracks and their merging and splitting processes are identified based on spatial and boundary changes of vortices across consecutive time steps. For each vortex with available AVHRR data, we generate Vortex-Centered Infrared (VCI, mentioned in the following text) images to identify corresponding cyclonic cloud features with a cyclone-detection deep-learning model. Finally, near-surface wind fields derived from QuikSCAT/ASCAT are matched to characterize cyclones' core wind speed. The algorithm workflow is outlined in Fig. 2, with methodological details provided in subsequent subsections.




Figure 2 The workflow diagram. In the diagram, all methodologies are enclosed in dashed circular outlines, while derived datasets are framed in solid rectangular boxes. The title of each swimlane denotes the data utilized by all methods within that swimlane.

# 3.1 Objective algorithm for identifying and tracking vortices

Sea-level pressure (Laffineur et al., 2014; Michel et al., 2018) and low-level relative vorticity (Day et al., 2018; Stoll et al., 2021; Watanabe et al., 2016; Zappa et al., 2014) are the two most common tracer variables for PMCs in reanalysis datasets. Existing studies demonstrate that high values of low-level relative vorticity, compared to sea-level lows which are susceptible to synoptic scale pressure fields, are more closely associated with actual cyclone positions and exhibit smaller biases in cyclone detection and intensity estimation (Stoll, 2022; Stoll et al., 2020; Zappa et al., 2014). Therefore, we apply an objective mesoscale vortices-tracking algorithm to the 850 hPa relative vorticity fields in ERA5 data to obtain

hourly-resolution vortex tracks. This algorithm was first proposed by Shimizu and Uyeda (2012) to track convective cells prone to merging and splitting, and has since been developed and improved for PMC tracking (Watanabe et al., 2016; Stoll et al., 2021). It specifically comprises two components: hourly vortices identification and connection of continuous time step vortices.

#### 3.1.1 Hourly vortices identification







When multiple vortices coexist within the same region of cyclonic shear flow, they often manifest as a contiguous positive vorticity zone in the vorticity field (hereafter referred to as an unpartitioned-vortex in the algorithm). The major challenge in vortex identification within vorticity fields is how to partition such regions (as exemplified in Fig. 3) into distinct isolated vortex regions.

Figure 3 (a) 850 hPa relative vorticity field obtained by ERA5 data. (b) AVHRR infrared imagery concurrent with the time step in (a). The shading represents 850 hPa relative vorticity smoothed over a uniform 60 km radius and local vorticity maxima are marked by green star symbols, while regions enclosed by solid black contours denote the unpartitioned-vortex zone.

First, a uniform 60 km smoothing radius is applied to hourly 850 hPa relative vorticity to disconnect weak continuity zones and eliminate minor perturbation maxima, which may arise from assimilation increments (Belmonte Rivas and Stoffelen, 2019). Subsequently, in the smoothed vorticity field, regions enclosed by closed contour lines exceeding a minimum threshold  $\zeta_{min0}$  are identified as unpartitioned vortices. Thereafter, each unpartitioned-vortex (e.g., the area within the thick black solid line in Fig. 4) is subjected to isolated vortex extraction via the following procedure:

Step 1: Identify local vorticity maxima exceeding the threshold  $\zeta_{max0}$ , designated as vortex peaks with relative vorticity values  $\zeta_{max}$  (e.g., in Fig. 4, three local vorticity maxima satisfy b > a > c). Contour lines (gray thin solid lines) are then drawn at  $10^{-6}$  s<sup>-1</sup> intervals. Subsequently, the outermost contour line enclosing each individual or combined peak (s) is identified as the valley-line (black thin solid lines, e.g.,  $\zeta_{min1} \approx \zeta_{min2} < \zeta_{min3} \approx \zeta_{min4}$  in Fig. 4). These valley-lines enable the separation of distinct vortex regions containing single or multiple peaks.

Step2: The isolation status of each vortex region is determined by assessing the relative disparity between each valley-line and its internal maximum peak. As illustrated in Fig. 4: peak a represents the strongest peak within its associated valley-line  $\zeta_{min4}$ , peak b corresponds to the maximum within two valley-line-enclosed areas  $\zeta_{min1}$  and  $\zeta_{min3}$ , and peak c is the dominant peak within its respective valley-line  $\zeta_{min2}$ .

The assessment proceeds systematically through vortex regions in descending order of their valley-line vorticity magnitude ( $\zeta_{min}$ ): for the maximum peak with relative vorticity value  $\zeta_{max}$  within the valley-line-enclosed vortex region, if the criterion ( $\zeta_{max} - \zeta_{min}$ )/ $\zeta_{max} > \gamma$  is satisfied (where  $\gamma$  denotes the isolation vortex threshold), the area centered on this peak and bounded by the valley-line is classified as an isolated vortex region. If a vortex region contains only one such isolated vortex region, the isolated vortex will be expanded to encompass the entire domain. (in Fig. 4, the vortex region enclosed by  $\zeta_{min4}$  associated with peak  $\alpha$  fails to meet the isolation criterion. Conversely, peaks  $\alpha$  and  $\alpha$  forming two distinct isolated vortex regions bounded by their respective valley-lines  $\zeta_{min1}$  and  $\zeta_{min2}$ ).

Step3: For all vortex points located within each unpartitioned-vortex but outside the isolated vortex regions, each point is assigned to the nearest isolated vortex based on geographical distance. Finally, all isolated vortices in the each unpartitioned-vortex region are mutually designated as adjacent vortices (e.g., vortices b and c), serving as inputs for subsequent analysis of merging or splitting events. The area of each vortex is defined by its corresponding allocated isolated vortex region.

Figure 4 Vortex identification algorithm example. The black thick solid lines  $\zeta_{min0}$  represent the unpartitioned-vortex border. The vorticity peaks a, b, and c are three detected local vorticity maxima within this unpartitioned-vortex. The thin black solid lines from  $\zeta_{min1}$  to  $\zeta_{min4}$  in Step 1 denote vortex valley-lines that divide single or multiple peak regions. After vortex isolation assessment in Step 2, the retained valley lines  $\zeta_{min1}$  and  $\zeta_{min2}$  for peaks b and c form the initial boundaries of their respective isolated vortices, while vortex a is classified as non-isolated, with its boundary shown as a dashed line. In Step 3, the pale pink regions outside the isolated vortices are further allocated to vortices b and c.

# 3.1.2 Connection of continuous time step vortices





Based on the results of hourly vortices identification, the introduction of steering wind is employed to estimate the movement of vortices. The steering wind is computed by averaging wind fields within a 450 km radius around the vortex center at 550 hPa, 700 hPa, and 850 hPa, which is statistically proven to have minimal bias (Yan et al., 2023). Specifically, for a vortex at a given time step, its ideal point after experiencing a time step under the steering wind influence is first calculated. A search radius of 180 km is then applied around this estimated location to facilitate vortex tracking in subsequent time steps. Subsequently, the (a) nearest neighbor principle or (b) maximum area overlap principle (as shown in Fig. 5) is applied to connect vortices between two consecutive time steps, when vortices exist within the

estimated region, the nearest vortex is connected; otherwise, the vortex with the largest area overlap within the region is selected for connection. Finally, if the distance between the centers of vortices to be connected in adjacent time steps exceeds 200 km and the vorticity of the vortex center at next time step is less than  $1.5 \times 10^{-4} \, \text{s}^{-1}$ , the connection is terminated to minimize spurious connections.

Figure 5 Schematics of continuous time step vortices connection

Additionally, If no spatially connectable vortices are identified in adjacent time steps, the vortex is classified as being terminate. Under the assumption of constant centroid positions during splitting and merging (Shimizu and Uyeda, 2012), if a vortex is contiguous to other vortices at its start (end) track point, it is considered to have been generated (terminated) via splitting (merging). As shown in Fig. 6, in two simplified vortex motion scenarios, vortex *b* begins splitting and merging at the t3 time step.

Figure 6 The schematic diagram illustrates two vortices splitting and merging processes. The t1 to t4 represent four consecutive time steps. The red/ blue arrow indicates the direction corresponding to the splitting/ merging process of two vortices. The colored regions and solid lines represent isolated vortex regions and their boundaries. Gray solid lines show contour lines of the 850 hPa relative vorticity field, and black solid lines indicate the unpartitioned-vortex boundaries. The blue dashed line indicates that the vortex b is not yet an isolated vortex at time t2.

## 3.1.3 Sensitivity experiments of vortex identification parameters






To evaluate the sensitivity of vortex identification parameters, we conducted three sensitivity experiments with the following configurations, each designed to test the impact of varying key thresholds  $\zeta_{max0}$  ( $\zeta_{min0}$ ) and  $\gamma$  on vortex detection:

- 1) Experiment a (lenient thresholds):  $\zeta_{max0} = 1.2 \times 10^{-4} \text{ s}^{-1}$ ,  $\zeta_{min0} = 1.0 \times 10^{-4} \text{ s}^{-1}$ ,  $\gamma = 0.15$ ;
- 2) Experiment b (intermediate thresholds):  $\zeta_{max0} = 1.2 \times 10^{-4} \text{ s}^{-1}$ ,  $\zeta_{min0} = 1.0 \times 10^{-4} \text{ s}^{-1}$ ,  $\gamma = 0.25$ ;
- 3) Experiment c (strict thresholds, following Stoll et al. 2021):  $\zeta_{max0} = 1.5 \times 10^{-4} \text{ s}^{-1}$ ,  $\zeta_{min0} = 1.2 \times 10^{-4} \text{ s}^{-1}$ ,  $\gamma = 0.25$

The influence of threshold variations on vortex detection characteristics was systematically evaluated by analyzing differences in the number of identified vortex tracks, their lifespans, and their

vorticity across the three experiments. As shown in Fig. 7, threshold adjustments predominantly affected vortices exhibiting maximum vorticity ( $\zeta_{trmax}$ ) less than  $2 \times 10^{-4}$  s<sup>-1</sup>. The principal findings are:

First, focusing on the impact of  $\zeta_{max0}$  (by comparing Experiment b, which uses a lenient  $\zeta_{max0}$ , with Experiment c, which uses a strict  $\zeta_{max0}$ ), we found that the lenient threshold in Experiment b captured an additional 8,077 weak-vorticity tracks (with  $\zeta_{trmax} 

Figure 7 Sensitivity analysis of vortex identification parameters across different maximum track vorticity groups: (a) number of identified tracks, (b) mean track lifetime.

## 3.2 Matching SLP minimum








While vortices often fail to produce closed isobars in SLP fields due to interference from background pressure gradients, their atmospheric influence can still be quantified through detectable SLP minima. Notably, certain polar lows originate within upper-level cold-core systems (known as "cold low types") frequently generate deep convection and produce substantial near-surface impacts (Rasmussen, 1981; Businger and Reed, 1989). To systematically capture SLP characteristics, the SLP field is first smoothed using Gaussian filtering with a radius of 50 km to suppress noise. Subsequently, the SLP minimum point located within a 150 km radius of the nearest vortex centroid is designated as the SLP center for that vortex.

#### 3.3 Detection and extraction of cyclonic cloud characteristics

Building upon the lenient vorticity identification criteria previously constructed, a substantial population of vortex tracks have been identified using reanalysis data, including not only cyclonic systems but also low-pressure troughs, and small-scale atmospheric disturbances. To assess whether these vortices represent PMCs, AVHRR infrared imagery is used for comparative validation. This process begins with temporal matching of satellite overpasses to vortex track timesteps, followed by generation of Vortex-Centered Infrared (VCI) images through linear interpolation of infrared data onto a geographically-referenced 801×801 grid coordinate with 2 km resolution, centered on each vortex center (Fig. 8c and Fig. 8d). The coordinate transformation employs the formulas:

$$lat(x,y) = \frac{y}{2\pi R} + vort_{lat}, x, y \in \{-800, -798, \dots, 798, 800\}$$
 (1)

$$lon(x,y) = \frac{x}{2\pi R \cdot cos(vort_{lat})} + vort_{lon}, x, y \in \{-800, -798, \dots, 798, 800\}$$
 (2)

The coordinate transformation utilizes  $vort_{lon}$  and  $vort_{lat}$  as the longitude and latitude of the original coordinate grid, corresponding to either the vortex center at the given timestep. This approach implements a conformal projection that provides a first-order approximation of geographic coordinates within the vicinity of the origin point.

The VCI images enable comprehensive analysis of cloud features within a 1600 km×1600 km domain centered on each tracked vortex position, providing an optimal spatial scale that captures the majority of PMCs while simultaneously accommodating larger-scale extratropical systems advected into Arctic regions. By transitioning from broad-scale satellite observations to these precisely localized domains, this imagery method significantly enhances the spatial correspondence between vorticity-derived tracks and cloud features, with particular sensitivity improvement for smaller-scale and shallower cyclones. Meanwhile, the georeferenced framework of VCI images provides two critical analytical capabilities: first, it enables direct quantification of cyclone dimensions through the standardized geographic grid; second, it allows precise measurement of positional discrepancies between observed cloud systems and modeled vortices through center-to-center displacement vectors. Furthermore, VCI images are also generated for two-time steps before the start and after the end of each vortex track. This allows us to capture the initial formation and dissipation stages of PMCs that are not adequately

Figure 8 Two examples of VCI image generation. For the two vortices shown in (a), the AVHRR IR image (b) reveals a polar low located to the east of vortex 1 and vortex 2. This polar low exists simultaneously in the VCI images centered on vortex 1 and vortex 2 (c, d). The shading in (a) represents 850 hPa relative vorticity smoothed over a uniform 60 km radius, with gray contour lines indicating sea-level pressure at 10 hPa intervals. The centers of vortex 1, vortex 2, and the polar low are respectively marked by green, red, and yellow stars.

Fig. 9 illustrates typical cyclonic cloud morphologies, the most common comma-shaped cloud structure is shown in Fig. 9a, where the head is typically composed of a tall and smooth cirrus shield surrounding a dark, nearly cloud-free center. Ripple-like wave patterns sometimes appear at the edge of the head, indicating significant wind shear within the cyclone. Fig. 9d presents the typical spiral cloud morphology, characterized by one or more convective cloud spiral bands encircling the circulation center. These spiral bands are occasionally predominantly composed of cellular clouds. Intermediate baroclinic forms illustrated in Fig. 9b and Fig. 9c represent transitional stages between comma and spiral types, sharing structural similarities with occluded extratropical cyclones but at reduced horizontal scales, and are consequently classified within the spiral category. The centers of comma cloud and spiral cloud configurations in our research were visually determined following Forbes and Lottes (1985), based on the characteristic curvature and convergence of cloud bands surrounding the circulation core as identified in satellite imagery. Additionally, the analytical framework of oriented bounding box is also introduced that provide quantitative measures of cyclone scale, with the long axis aligned parallel to the tail cloud

band and the short axis tangent to the cloud head. While conventional approaches estimate cyclone size using the mean axis length (Smirnova et al., 2015), this dataset provides separate measurements of both axes to account for potential overestimation caused by the connection of tail cloud band of cyclones and long cloud bands of mesoscale-front, thereby enabling researchers to make more precise assessments of true cloud coverage dimensions.




Figure 9 Different cyclonic cloud morphologies in four VCI images: (a) comma-shaped cloud; (b), (c) and (d) spiral clouds. The yellow/blue bounding boxes and stars respectively denote the oriented bounding boxes and center positions of comma-shaped/spiral cyclones.

To extract such cyclonic cloud features corresponding to vortices from the vast collection of VCI images, the YOLO (You Only Look Once) object detection algorithm is employed to automate this process. Object detection is a computer vision task that uses neural networks to locate and classify objects within images. The YOLO series of algorithms (Redmon et al., 2016), characterized by high efficiency and accuracy, has become prominent in real-time object detection tasks across various fields. In this track dataset construction, the YOLOv8 framework (Jocher et al., 2023) is adopted to automatically extract cyclonic cloud morphology features, including cloud type classification (spiral cloud or commashaped cloud), center coordinates, and an oriented bounding box enclosing the cyclone. The YOLOv8-obb-pose model is configured using the YOLOv8 model framework, which combines oriented object detection (obb) and keypoint detection (pose). Specifically, a branch for keypoint prediction is added to the decoupled head module of the YOLOv8-obb model. This enables the new YOLOv8-obb-pose model to simultaneously perform automatic detection of cyclone type, center position, and oriented bounding box. The network architecture of the YOLOv8-obb-pose model comprises three main components: Backbone for multi-dimensional feature extraction, Neck for enabling multiscale feature fusion, and

Head for extracting cyclone type, center coordinates, and oriented bounding box parameters (e.g., length, orientation). As shown in Fig. 10, the YOLOv8-obb-pose model successfully detects two spiral clouds (Fig. 10a) and two comma-shaped clouds (Fig. 10b) in VCI images, with oriented bounding boxes, cyclone type and center points marked.

During the model training process, we first construct a manually annotated dataset to train the YOLOv8-obb-pose model. To ensure prediction stability, particular emphasis is placed on maintaining consistent oriented bounding box annotations and center point positions across similar evolutionary phases of cyclonic cloud morphologies. To optimize the trade-off between detection efficiency and accuracy, we implement an iterative training protocol involving successive cycles of prediction, manual correction, and retraining using VCI images. As detailed in Table S1, the model achieves competitive performance metrics on the validation set following this optimization process. The final YOLOv8-obb-pose implementation demonstrates robust capabilities in both cyclone detection and center localization tasks, satisfying requirements for practical applications.

For each detected cyclone, the center coordinates and the four vertices of the oriented bounding box are converted back to geodetic coordinates using the inverse of Eq. (1) and (2). The lengths of the four sides of the bounding box are calculated using the haversine formula, with the cyclone's length (width) defined as the mean size of the two long (short) sides of the rectangle. The geographic coordinates of the cyclone center are then used for subsequent matching with vortex centers.

# (a) detected spiral clouds







# (b) detected comma clouds

Figure 10 Examples of cyclonic cloud detection using the YOLOv8-obb-pose model: (a) two spiral clouds detected in a VCI image and (b) two comma-shaped clouds detected in a VCI image. The oriented bounding boxes for spiral clouds are shown in purple, and for comma-shaped clouds in blue. The centers of the cyclones are marked with green points. The cyclone type and detection confidence are displayed above each bounding box.

# 3.4 Validation of the vortex tracks

Each series of VCI images based on vortex track provides spatiotemporal neighboring local infrared cloud imagery that follows the vortex's movement. After extracting cyclonic features from VCI images, whether a vortex track corresponds to a cyclone evolution process is determined by proximity matching

between the cyclone center detected in each VCI image and the vortex center. The following steps ensure that each VCI image only retains a cyclone uniquely matched to a vortex track point:

I Uniqueness: As illustrated in Fig. 8, spatially proximate vortices in reanalysis data can result in multiple detections of the same cyclone across corresponding VCI images. To remove duplicate records, we implement a selection criterion: for any cluster of detections from the same AVHRR infrared scan (with cyclone centers 

Figure 11 (a) A matched vortex track and cyclone track and (b) partial corresponding VCI images. For (a), blue solid line represents the vortex track at hourly resolution, while grey solid line with green points depicts the cyclone track points formed in VCI images that correspond one-to-one with vortex points. The color of the track points indicates the magnitude of relative vorticity at each vortex point. For (b), the cyclone develops sequentially from left to right and top to bottom, with scan intervals between images approximately six hours apart.

## 3.5 Matching cyclone-related max wind and environmental near near-surface wind

When cyclonic cloud features are identified in VCI imagery, near-surface wind speeds over the ocean are matched to assess cyclone intensity. Based on established criteria (Rasmussen and Turner, 2003), PLs are generally associated with high near-surface wind speeds exceeding 15 m s<sup>-1</sup> (gale force), concentrated in narrow cloud bands connected to the eye wall or intense convective regions surrounding the center. In contrast, weaker PMCs often do not penetrate the temperature inversion above the marine mixed layer, resulting in lower near-surface wind speeds (Noer et al., 2011). In this study, near-surface wind speed matching is performed using ASCAT/QuikSCAT data selected when the time difference from the VCI image is within 30 minutes. This tolerance is considered acceptable given that most PLs move at speeds below 13 m s<sup>-1</sup> (Rojo et al., 2015; Smirnova et al., 2015), making the associated representative error negligible. To estimate the maximum wind speed associated with the cyclone core, a cloud-scale-based search radius is applied. The search radius is defined as the distance from the cyclone center to the

nearest short edge of its oriented bounding box. This confines the wind search to the high-wind region near the cyclone's core, with the maximum value within this area taken as the system's maximum wind speed.







It is important to recognize that scatterometer wind speeds may not always reflect cyclone-induced circulation and could include contributions from large-scale advective wind. Some PMCs occurring during cold air outbreaks may exhibit wind speed maxima surpassing 15 m s<sup>-1</sup> due to background environmental wind advection. To prevent misclassifying such systems as PLs, careful subjective analysis has traditionally been applied (Wilhelmsen, 1985). This highlights that what is retrieved from scatterometer wind measurements may not always reflect cyclone-induced circulation, but could also include contributions from large-scale advective winds. By using the spatial derivatives from scatterometer wind vector fields, vortical structures or divergent flows near the surface associated with PLs/PMCs may become easily visible (King et al., 2022). For instance, Fig. 12a illustrates a system with a well-defined cyclonic circulation where the high wind speeds at its head are clearly associated with the cyclone itself. The fine-scale and complex structure of the corresponding vorticity field exhibits a strong and organized vorticity signature coincident with the cloud vortex, confirming the presence of an intense mesoscale vortex and a trailing shear line. In contrast, Fig. 12b shows a case where the wind field is largely straight and convergent in the ambient flow, accompanied by only a weak vorticity signal (1×10<sup>-4</sup> s<sup>-1</sup>) localized near the cloud eye and lacking any broader organized cyclonic structure, suggesting that the surface circulation appears to be either not yet formed or obscured. Due to technical constraints, additional parameters such as vorticity and divergence are not provided alongside wind speed. Nevertheless, they retain substantial application potential, as evidenced by the vorticity structures revealed in Fig. 12, which demonstrate the value of scatterometer spatial derivatives in elucidating the complex dynamical features of mesoscale systems.

Figure 12 VCI images overlaid with near-surface wind speeds for cyclones exhibiting strong (a) and weak (b) cyclonic near-surface wind patterns. Color shading represents QuickSCAT-measured 10m near-surface wind speeds, with green arrows indicating corresponding wind vectors. Yellow borders denote the cyclones' bounding oriented box. Blue circular border represents the search range. Yellow and red stars indicate the

cyclone center and maximum wind speed point locations. The vorticity calculated from the wind fields is shown as white-to-red contours, with units of  $10^{-4}$  s<sup>-1</sup>.

# 4 Results and discussion









Our analysis began by applying a vortex tracking algorithm to reanalysis data, which identified 59,975 vortex tracks. Validation against VCI imagery confirmed 1,110 cyclone-related vortex tracks, encompassing 16,001 distinct cyclone cloud features. Subsequent analysis of surface wind speed characteristics revealed 4,472 instances with measurable wind patterns, among which 794 tracks exhibited maximum wind speeds exceeding the 15 m s<sup>-1</sup> threshold. These validated 1,110 vortex tracks, along with their corresponding remote-sensing images, form the IMPMCT track dataset. The accuracy of IMPMCT was rigorously evaluated through comprehensive comparisons with existing track datasets derived from manual identification and reanalysis products.

First, to validate the accuracy of the vortex track datasets obtained from the vortices tracking algorithm, they are compared with the manually identified PL lists published by Noer et al. (2011), Rojo et al. (2019), and the objectively derived PL track datasets from reanalysis data by Stoll (2022). All reference datasets are spatially and temporally co-located with our derived tracks, retaining only those persisting for ≥3 hours. We applied the following matching criteria: a vortex track is considered matched with a PL track if more than 50 % of temporally coincident track points (within ±1 hour) fall within a 150 km radius (applying an 80 % threshold for Stoll's dataset). To avoid spurious matches of short-lived spurious tracks, only vortex tracks with lifespans exceeding 60% of the corresponding reference PL track's duration were included. A single vortex track was permitted to match multiple PL tracks from reference datasets, provided that these PL tracks did not overlap temporally and each was uniquely paired with its nearest vortex track. As presented in Table 2, the validation results demonstrate strong agreement with Stoll's dataset, confirming the robustness of our vortex tracking algorithm. Moreover we achieve higher matching rates with manual PL lists by using lower vortex identification thresholds, which further underscores the improved capability of ERA5 reanalysis data in representing PL characteristics. Additional validation using tracks from the sensitivity experiment (Sect. 3.1) revealed a critical insight: vortex tracks derived under lenient thresholds consistently produced higher matching rates when compared against established PL datasets (Table S2). This suggests that some PLs exhibit weaker vorticity signals in the lower atmosphere, highlighting intrinsic intensity diversity that stricter thresholds may fail to capture.

To further investigate the mismatches between reanalysis-derived tracks and existing PL datasets, we conducted a nearest-point matching analysis (Table 2). A match was considered successful when a PL center from any reference dataset had at least one temporally coincident vortex center within a 120 km radius (60 km for the Stoll dataset). Track-level mismatches were found to originate primarily from these point-level discrepancies. The variation across datasets can be largely attributed to methodological differences: while the Noer list derives from numerically modeled and AVHRR-assimilated hourly positions (typical of operational forecasting systems), the Rojo list relies on direct AVHRR identification at irregular temporal intervals, leading to greater deviation from ERA5 representations. Furthermore, the Rojo compilation includes numerous secondary PL centers, which are features inherently less resolved

by reanalysis data (Stoll, 2022), whereas Noer focuses primarily on dominant PLs of operational significance. This distinction is clearly reflected in our results: major PL centers (n = 2,527) showed an 80% matching rate, compared to only 54% for secondary centers (n = 1,115), thereby lowering the overall match rate for the Rojo dataset.

For the Stoll dataset, we also computed a vortex matching rate (Table 2), defined as the proportion of Stoll centers falling within the spatial extent of the nearest co-temporal vortex. This measure helps account for positional discrepancies caused by misalignment of vorticity peaks, which appear to stem from differences in smoothing techniques (see Fig. S1). Our algorithm applies stronger uniform smoothing compared to Stoll's approach, explaining why more lenient identification thresholds improve track matching with Stoll's dataset. This finding offers valuable insight for algorithm application: although the algorithm is not highly sensitive to the specific input vorticity fields, provided their grid spacing is sufficient to capture mesoscale vortices, the choice of smoothing method significantly influences identification outcomes, alongside the threshold parameters examined in the sensitivity experiments (Sect. 3.1.1). The smoothing strategy should be tailored to the assimilation noise and effective resolution of the input vorticity field. For example, Gaussian smoothing may be better suited for model data with lower noise levels, as it better preserves the spatial coherence of vortex cores.

Table 2 the matching rate of the reanalysis-based track dataset for IMPMCT generation compared to other PL track datasets.

| PL tracks | Time<br>period | Tracks<br>in Nordic<br>Sea (>3hr) | Track<br>matched<br>fraction(%) | Points | Nearest points<br>matched<br>fraction(%) | Vortex<br>matched<br>fraction |
|-----------|----------------|-----------------------------------|---------------------------------|--------|------------------------------------------|-------------------------------|
| Noer      | 2002-2011      | 114                               | 87.72                           | 1670   | 85                                       | -                             |
| Rojo      | 2000-2019      | 370                               | 69.73                           | 3642   | 71                                       | -                             |
| Stoll     | 2000-2020      | 3179                              | 93.68                           | 75650  | 93                                       | 99                            |

After excluding vortex tracks with over 60% land coverage (resulting in an approximately 20% reduction), 47,167 tracks remained eligible for AVHRR matching. The matching procedure required: (1) complete spatial coverage within a 200-km radius for individual vortex points, and (2) at least two temporally matched points within ±3 hours of peak vorticity, along with a minimum of six matched points over the track's lifetime. Figure 13 presents the matching statistics for the winter months (November to April): on average, 43% of points and 61% of tracks were successfully matched. However, only about 3% of the matched tracks were ultimately incorporated into the IMPMCT dataset. This low inclusion rate can be attributed to several factors: frequent cloud obstruction, limitations in cloud–ice contrast, temporal resolution constraints, and inherent detection methodology (e.g., the higher inclusion rate in 2001 reflects meticulous manual identification, whereas the lower rate in 2023 resulted from incidental post-publication discoveries). Importantly, the proportion of cyclones in IMPMCT likely underestimates the true prevalence of polar mesoscale cyclones (PMCs), as many systems with low cloud cover lack discernible vortex structures. In cases where AVHRR data are unavailable, an alternative approach using

hourly wind field data calibrated with scatterometer measurements may provide a more robust method for validating ERA5-derived vortex tracks (Furevik et al., 2015).






Figure 13 Annual winter (November-April) time series: (a) ERA5-derived vortex points (green), available AVHRR files (red), and AVHRR-matched vortex points (blue), (b) ratio of AVHRR-matched vortex tracks to ERA5-derived tracks (yellow), and ratio of IMPMCT tracks to AVHRR-match tracks (purple). Note: Bars represent distinct categories (not stacked)

We further assess the reliability of vortex properties in IMPMCT by comparing three key parameters (850 hPa relative vorticity, SLP minima, and vortex equivalent diameter), with the corresponding values from Stoll's dataset, in addition to evaluating the spatial distance between vortex centers. From this comparison, 638 matched tracks were identified between IMPMCT and Stoll's dataset. As shown in Fig. 14a, among the matched tracks, 90 % of vortex points remain within 50 km of each other at the same time step. The mean absolute differences of the three vortex properties at these proximate track points remain small:  $1.11 \times 10^{-5}$  s<sup>-1</sup> for relative vorticity, 0.43 hPa for sea-level pressure, and 22.79 km for vortex equivalent diameter. Furthermore, these property discrepancies exhibit a positive correlation with separation distance, suggesting that differences between IMPMCT and Stoll's tracks primarily arise from their respective identification thresholds.

To demonstrate that these discrepancies reflect divergent tracking methodologies rather than detection errors, we calculated the standard deviation of each vortex property over three consecutive time steps for every track and then averaged these values across each track. Low amplitude in these local variations implies consistent feature identification by a given method. Figures 14b—d present the track-averaged local standard deviations of the three properties for both datasets. Importantly, the magnitudes

of these short-term variabilities are generally comparable between IMPMCT and Stoll's tracks. This consistency indicates that the increasing property differences at larger separations stem from intrinsic peak misalignments due to differing detection logics, rather than fundamental errors in either tracking approach. In fact, the IMPMCT dataset often exhibits slightly smoother variability, which is consistent with its specific algorithmic configuration.

Figure 14 Distribution of differences in three vortex properties and their track-averaged local standard deviations at co-located hourly track points between matched IMPMCT and Stoll tracks. The boxplot in (a) shows property differences as a function of spatial deviation distance between matched track points. The red numbers above the x-axis indicate the count of track point pairs in each distance bin. Each boxplot's y-axis scale corresponds to the color of its respective property (green: relative vorticity, blue: sea-level pressure, red: vortex diameter). Frequency histograms and fitted curves of track-averaged local standard deviations for the three properties are displayed in (b) relative vorticity, (c) sea-level pressure, and (d) vortex diameter.

IMPMCT uses hourly-resolution vortex tracks from reanalysis data as a basis for cyclone tracks. The correspondence between vortex and cyclone tracks is established exclusively via continuous spatiotemporal matching of their respective centers. To ensure the accuracy of this correspondence, we perform subjective validation to confirm that each cyclone track does not incorporate irrelevant cyclonic processes. Notably, while the average matching distance between vortex and cyclone tracks is constrained within 150 km, approximately 95 % of track pairs have average matching distances below 100 km (as shown in Fig. 15), demonstrating strong consistency between cyclone and vortex tracks.

Figure 15 Probability distribution of distances between matched cyclone-vortex points (green) and track-average distances (blue).

The cyclone properties in IMPMCT include cyclone scales and maximum core near-surface wind speeds. These properties are validated through comparison with the Rojo list. For scale validation, we compare the diameter from the Rojo list with the approximate cyclone scale in the IMPMCT dataset (calculated as the average of cyclone width and length). We matched cyclone tracks between IMPMCT and Rojo list based on the following criteria: the nearest cyclone centers are matched if their distance is less than 120 km and their overpass times fell within 60 minutes of each other. A cyclone track pair was deemed matched if more than 50% of the points in a Rojo track were matched. Using this approach, 1424 cyclone centers from the Rojo list (corresponding to 139 distinct tracks) were matched to tracks in IMPMCT. It is worth noting that although the maximum permitted matching distance was 120 km, the 90th percentile of all actual matching distances was only 56 km. This indicates that cyclone center identification remained consistent even when exact temporal alignment was not achieved.

Comparisons of cyclone cloud scale and maximum wind speeds between the matched time periods are shown in Fig. 16. When cyclone center identification errors are small, the discrepancies in diameter relative to the Rojo list arise not only from methodological differences in measurement, but also significantly from subjective interpretation. The frequent presence of frontal cloud bands associated with cyclones makes consistent measurement of the long axis highly subjective. Moreover, when a cyclone is adjacent to other cloud systems, its boundaries often become ambiguous, leading to variability in extent estimation. Therefore, a standard deviation of up to 120 km in diameter is still considered acceptable. Furthermore, as the dataset includes corresponding remote sensing images, users can readily examine the visual context of each cyclone and adjust the properties according to their specific research needs.





Figure 16 Frequency distribution of bias in (a) Track-max near-surface wind speed and (b) diameter between matched cyclones in the Rojo and IMPMCT datasets (Rojo minus IMPMCT). The cyclone diameter in IMPMCT is calculated as the average of the width and length of the bounding box enclosing the cyclone.

To statistically evaluate the agreement between IMPMCT and the reference datasets (Stoll, 2022, and Rojo et al.), we applied Bland–Altman analysis (Bland and Altman, 1999). This method quantifies the agreement between two measurement techniques by estimating the mean difference (bias) and the 95% limits of agreement (LoA), defined as the mean difference  $\pm$  1.96 standard deviations of the differences. A summary of the Bland–Altman results for key vortex and cyclone properties is provided in Table 3, while the corresponding plots of differences versus averages are included in Supplementary Fig. S2. As indicated in Table 3, vortex properties derived from ERA5 reanalysis show a small systematic bias relative to the other datasets, which is likely due to differences in computational algorithms or processing workflows. Importantly, the Bland–Altman results demonstrate strong agreement between the datasets: approximately 94% of the differences for each property fall within the respective 95% limits of agreement (final column of Table 3), supporting the overall consistency and reliability of IMPMCT

Table 3 Property difference between IMPMCT and other PLs list

| Property                                                       | Matched number | Mean<br>Difference | Standard Deviation of Differences | % Points within LoA |
|----------------------------------------------------------------|----------------|--------------------|-----------------------------------|---------------------|
| 850 hPa relative vorticity (10 <sup>-5</sup> s <sup>-1</sup> ) | 20294          | 0.6                | 2.1                               | 95.1                |
| SLP (hPa)                                                      | 13929          | 0.3                | 0.8                               | 95.7                |
| vortex equivalent diameter (km)                                | 20294          | -6.8               | 39.2                              | 93.7                |
| track-max near-surface wind speed (m s <sup>-1</sup> )         | 51             | -1.07              | 5.0                               | 94.1                |
| cyclone cloud diameter (km)                                    | 1145           | 8.8                | 120                               | 94.5                |

For most newly identified mesoscale cyclones not documented in existing PL databases, direct validation can be performed by applying objectively derived identification thresholds from previous studies to independently verify three essential characteristics: polar origin, mesoscale size, and cyclonic intensity:

1) Polar-front criterion: As PMCs are defined as mesoscale cyclones forming north of the polar front (Rasmussen and Turner, 2003), we employ two indicators to distinguish polar air masses from extratropical air masses: tropopause potential temperature (θ<sub>trop</sub>) and the maximum poleward value of 200 hPa wind speed (U<sub>200,p</sub>). For each cyclone, we compute the track-averaged θ<sub>trop</sub> averaged within a 250 km radius of the cyclone center and the track-averaged

 $U_{200,p}$  within a longitudinal band of  $\pm 1.0^{\circ}$  great-circle distance. Following Stoll (2022),  $\theta_{trop}$  < 300.8 K is used to identify polar air mass origin. This threshold effectively distinguishes PLs from extratropical cyclones, retaining 76% of systems across subjective archives while capturing 90% of known PLs. Han and Ullrich (2025) employed U200,p < 25 m s<sup>-1</sup> to position PLs north of the polar jet, achieving an approximately 80% hit rate for PL classification with a miss rate of only 11.9%.







- 2) Mesoscale-size criterion: Vortex radius, derived from the vorticity field, is used to exclude extratropical cyclones penetrating polar regions and large-scale frontal structures. In Stoll (2022), a maximum vortex diameter of 430 km (representing the 90th percentile across all PL lists) was applied, excluding approximately 24% of non-PL vortices. As we employ the same vorticity boundary threshold (1.0×10<sup>-4</sup> s<sup>-1</sup>) for vortex definition, this criterion remains valid for our dataset.
- 3) Cyclonic intensity criterion: A robust measure of mesoscale cyclone intensity is the pressure anomaly ( $p_{def}$ ), defined as the difference between the SLP minima and the mean SLP within a 110 km radius ( $p_{def} = \overline{SLP}_{110km} SLP_{min}$ ). Stoll (2018) demonstrated that high  $p_{def}$  values (with 90% of PLs exceeding 0.4 hPa) highlight the anomalous intensity of the local low-pressure centre relative to its environment, signifying a steep pressure gradient near the core, indicative of small, deep low-pressure systems typical of PLs. We calculate the maximum  $p_{def}$  based on the SLP centre for each vortex track. For tracks where no SLP centre is identified,  $p_{def}$  is set to 0.

All discriminatory features for IMPMCT tracks are computed from ERA5 data. The quantiles of these features and the proportion of tracks meeting each criterion are presented in Table 4. Notably, 88.4% of tracks satisfy the polar-front criterion, 90% meet the mesoscale criterion, and 84% fulfill the cyclonic intensity criterion. It should be noted that these thresholds were originally developed specifically for the PLs. For the broader spectrum of PMCs, the thresholds for  $\theta_{trop}$  and  $p_{def}$  are inherently stricter, as they reflect the conditions of cold-air outbreaks and the stronger destructive potential typically associated with PLs. Nevertheless, the vast majority of tracks in the IMPMCT dataset satisfy these criteria, supporting their robustness as mesoscale cyclone tracks.

Table 4 Quantiles of discriminatory features and proportion of IMPMCT tracks meeting validation criteria.

| criterion                                                     | Track feature                        | percentage |       |       | Proportion                |
|---------------------------------------------------------------|--------------------------------------|------------|-------|-------|---------------------------|
| Criterion                                                     |                                      | 50%        | 75%   | 90%   | meeting the criterion (%) |
| Polar front                                                   | $\theta_{\text{trop}}\left(K\right)$ | 298.9      | 304.1 | 310.0 |                           |
| $\theta_{trop}$ < 301 K or $U_{200,p}$ < 25 m s <sup>-1</sup> | $U_{200,p}$ (m s <sup>-1</sup> )     | 18.4       | 23.7  | 29.7  | 88.4                      |
| Mesoscale<br>r< 215 km                                        | r (km)                               | 137.1      | 176.9 | 213.5 | 90.6                      |
| <b>Cyclonic</b> p <sub>def</sub> > 0.4 hPa                    | $p_{def}(hPa)$                       | 1.4        | 2.3   | 3.2   | 84.1                      |

The comprehensiveness of the dataset is constrained by the cyclone representation capabilities of

ERA5 reanalysis and the availability of remote sensing data. Since the number of in-orbit satellites carrying the AVHRR sensor peaked around 2013, the IMPMCT track dataset includes the highest number of tracks during this period. Additionally, due to the use of more lenient identification thresholds, IMPMCT tracks typically include longer life compared to the Stoll dataset. The extended portions of these tracks may include: weak vorticity periods during the early/late stages of cyclone development or the vortices pass over land/sea-ice, or redevelopment processes of vortices after interacting with blocked extropical cyclones or frontal zones. If users require only the core development phases of tracks, they should select segments based on vortex properties or cyclone images that represent the system's core development. The dataset also includes some tracks with high vorticity at their start/end points, which may arise from splitting/merging events or jumps of the vortex center position during tracking. It is noteworthy that while this study demonstrates ERA5 reanalysis data's enhanced capability in capturing PMCs and PLs, it does not reflect ERA5's predictive skill for such systems. This predictive capability should be evaluated by testing ERA5 background states in characterizing PLs/PMCs, thereby isolating the influence of real-time assimilated data—particularly scatterometer measurements (Furevik et al., 2015).

The dataset does not explicitly distinguish between PMCs and PLs due to the time-sparse wind speed data, particularly when the cyclone's wind speed at a given time step falls below the 15 m s<sup>-1</sup> threshold. In such cases, it is difficult to determine whether the cyclone is a PMC or merely in a weaker phase of a PL. In such cases, a more reliable validation method may be provided by the hourly biascorrected sea surface wind product from the E.U. Copernicus Marine Service Information (CMEMS, https://doi.org/10.48670/moi-00185). Such product systematically corrects ECMWF ERA5 model fields using scatterometer observations to reduce persistent biases and includes uncertainty estimates. Furthermore, the L3 scatterometer products available through CMEMS, which contain the spatial derivatives of the wind vector fields (vorticity and divergence), offer a more direct characterization of the dynamical core of mesoscale systems. These observed fields hold significant potential for refining objective identification criteria, moving beyond a reliance on wind speed thresholds alone. Due to the low resolution of AVHRR infrared images at scan edges, a significant portion of VCI images appear blurred. However, these images are retained as long as cyclonic features remain recognizable, prioritizing the preservation of high temporal resolution for cyclone track records. Additionally, while the YOLOv8obb-pose model facilitates detection and feature extraction of cyclonic cloud characteristics in VCI images, the process still involves subjective steps to ensure continuity in cyclone features (e.g., size, type, and position). This implies that objective methods for constructing multi-parameter PMC track datasets remain underdeveloped. Consequently, cyclone-evolution-aware deep-learning tracking algorithms could further enhance the efficiency of track construction.

#### 5 Code and data availability









The IMPMCT dataset described in this paper is freely accessible on Zenodo via the following link: <a href="https://doi.org/10.5281/zenodo.17142448">https://doi.org/10.5281/zenodo.17142448</a> (Fang et al., 2025), accompanied by comprehensive documentation. All code is developed in Python and stored at: <a href="https://github.com/thebluewind/IMPMCT">https://github.com/thebluewind/IMPMCT</a>.

#### **6 Conclusion**







The Integrated Multi-source Polar Mesoscale Cyclone Track (IMPMCT) dataset represents a major advancement in the study of polar mesoscale cyclonic systems. By integrating ERA5 reanalysis, AVHRR infrared imagery, and QuikSCAT/ASCAT wind data, this dataset provides a comprehensive record of 1110 vortex tracks, 16,001 cyclonic cloud features, and 4472 wind speed observations across the Nordic Seas (2001-2024). This integrated approach overcomes key limitations of previous single-source datasets by enhancing detection sensitivity for weaker polar mesoscale cyclones (PMCs), capturing complete lifecycle evolution from genesis to dissipation, and providing simultaneous cloud morphology and wind fields observations. Rigorous validation against established datasets (Stoll, 2022 and Rojo et al. , 2019) confirms IMPMCT's accuracy, demonstrating 90 % spatial consistency with track points cyclone centers alignments within 50 km (60 km for cyclone centers) and minimal parameter discrepancies including a 1.11 × 10<sup>-5</sup> s<sup>-1</sup> mean absolute difference in relative vorticity and 0.43 hPa mean absolute difference in sea-level pressure.

The IMPMCT dataset serves as a critical benchmark for evaluating high-latitude numerical weather prediction model performance, while simultaneously functioning as a unique case library for comparative studies of PLs and PMCs concerning their formation mechanisms, intensity thresholds, and sea-ice interaction dynamics. Furthermore, it constitutes an essential resource for enhancing polar maritime hazard forecasting. The repository of cyclone cloud morphology facilitates automated identification of model-undetected systems. This is enabled by advanced deep learning frameworks, enabling systematic evaluation of model representation fidelity for PLs/PMCs.

## **Author contributions**

RF conceived the experimental design and authored the manuscript. WG contributed to refining the methodologies. XL and HD conducted the research investigations and managed data collection. ZC and CZ contributed to the interpretation of the results. JD and LL provided critical guidance, reviewed, and revised the initial draft. All the authors contributed to the discussions and paper revision.

#### **Competing interests**

The contact author has declared that none of the authors has any competing interests.

## Acknowledgements

The work has been jointly financially by the project of National Key R&D Program of China (Project 2021YFC2802501) and NSF of China (No. 42476205).

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
