# Peer review of "IMPMCT: a dataset of Integrated Multi-source Polar Mesoscale Cyclone Tracks in the Nordic Seas"

_Earth System Science Data, 2025_

## Referee Comment (RC3)

**IMPMCT: a dataset of Integrated Multi-source Polar Meso-Cyclone Tracks**

Runzhuo Fang et al.

**General**

The manuscript describes a great data set and a laudable effort to construct such data base of PL and MPC tracks based on ERA5 and satellite data.

However, the characteristics and hence value of the data set is scientifically unclear. For existing similar track data sets, it is investigated how these are matched. It occurs that only a marginal set of points in the data base is characterized in the manuscript by these existing sets. Moreover, these appear the easiest tracks to capture, hence the value of most of the tracks remains unclear.

This is associated with the fact that I miss a critical scientific assessment of the tracks generated. The manuscript appears subjective, rather than rigorous.

There are ways to verify PL and MPC tracks with observations of atmospheric dynamics, in particular wind scatterometers. The use of scatterometers in this manuscript is rather unclear from a dynamic perspective and poor.

In the least, the manuscript should be scientifically clarified and the pros and cons of the methodology better stipulated. In addition, a section on future work appears appropriate as much remains unclear in my interpretation of the manuscript.

Detailed suggestions are provided below.

**Detailed suggestions**

95: These images are not so clear. In a): Could a PMC also be in (8,74), (36,77) or (36,77)? Why not? In b): Could the PL also be in (34,76)? Why not?

96: The ERA5 grid distance is 31 km, hence good dynamical representation will at most be 150 to 300 km following typical dynamical closure procedures. Is that good enough for PL/PMCs?

109: Belmonte Rivas and Stoffelen also suggest some other reasons for poor PL/PMC representation in ERA5: lack of transient variability, lack of divergence, lack of resolution; it appears of interest to mention these aspects.

113: Having looked at many collocated IR and scatterometer wind vector fields (e.g., here below), I have some problem with the terminology "cyclonic cloud feature". Cyclonic cloud features might occur due to closed surface circulation (cyclone definition) indeed, while wind shear conditions may also generate clouds in circles shapes on the mesoscales.

Moreover, a cyclone may also exist in an abundance or lack of clouds in which a cyclone is not recognized in an IR image. In the image below (from today) circular cloud patterns are present on the left hand side, while the streamlines of the vector winds do not coincide with the cloud streaks. On the other hand, a cyclonic wind feature appears on the right side of the plot, but where high clouds cover the wind structure below. This is today's example, while examples of apparent IR cloud mismatch with ocean vector winds occur almost every day on this site, in particular at high latitudes.

**Oceansat-3: 20250724 05:30Z lat lon: -64.0 -60.0 IR: 05:30**

From https://scatterometer.knmi.nl/tile_prod/index.php with description:

[Figure]

This picture shows the scatterometer winds (in arrows, flags or all ambiguities), with an infrared satellite image (from METEOSAT, GOES or Himawari) and numerical weather prediction model forecast winds from ECMWF in green arrows or flags. These model winds are valid at the time of observation. A wind flag is represented by barbs and solid pennants, a full barb representing a wind speed of 5 m/s, a half barb representing a wind speed of 2.5 m/s, and a pennant representing a wind speed of 25 m/s. A calm indicator circle is plotted if the wind speed is less than 0.5 m/s. The scatterometer winds are coloured according to the Beaufort scale, winds up to 5 Bft. (10.7 m/s) are in red, winds as of 6 Bft. are coloured as shown in the legend below the picture. A black arrow or flag indicates that the KNMI QC flag is set, such winds are likely to be unreliable but they may provide extra information to experienced users.

The ambiguity plots show up to 4 wind solutions that are input to ambiguity removal. The winds are shown as arrows without head, i.e., they point to the direction where the wind is blowing to. For the ambiguity plots a different wind speed colour scale is used. Infrared imagery and model winds are not shown in these plots. The winds having the KNMI QC flag set are coloured according to the colour scale but they are indicated with a black dot.

The exact data acquisition time is plotted in red next to the satellite swath.

The coloured dots give the value of the Maximum Likelihood Estimator (MLE) which indicates how well an observation fits to the Geophysical Model Function. High MLE values usually indicate high spatial wind variability or rain presence in the Wind Vector Cell.

170: remove "resolution"; Skamarock (2004) defines effective resolution as 5-10 times the grid distance of an atmospheric circulation model, due to the necessary dynamical closure for numerical stability of the model.

172: Note that in particular the initiation of PMCs and PLs in ERA5 is brought by wind scatterometers as can be observed in time sequences at https://scatterometer.knmi.nl/tile_prod/index.php. Hence ERA5 PMCs/PLs may be biased to the availability of the satellite data used, which could be problematic in time series analyses of PMCs/PLs. As readers may not be generally aware of this dependency, it is better to state it.

173: with a spatial resolution -> on a spatial grid

182: To refer to scatterometer accuracy, one may use Vogelzang and Stoffelen (2022).

192: ASCAT-A, -B and -C have been operational since 2007.

197: with stable spatiotemporal resolution -> exploiting the evolving global observing system ; I.e., not necessarily of stable spatiotemporal resolution effectively, since depending on the initialization of small scales by observations, when available.

208: Scatterometers measure the surface wind vector field and hence curl and divergence. See, e.g., Belmonte Rivas and Stoffelen (2019). King et al. (2022) found that tropical divergence as measured by scatterometers is closely related to moist convection. Similarly, one would expect that cyclonic disturbances are very well depicted in curl and divergence. These are furthermore available at https://data.marine.copernicus.eu/product/WIND_GLO_PHY_L4_MY_012_006/description. It also provides hourly corrected ERA5 wind variables for reference.

Why not put them in the database? They provide a stable reference over time as each instrument product does not change over time.

232: The vorticity field appears noisy as I understand the text. Nevertheless, no observations exist to initialize 4D dynamical structures well on scales below 100 km over the ocean, hence 60-km filtering may not be too problematic. The noise may be due to the fact that you use analyses, rather than more consistent dynamical model fields, i.e., background (first guess) ERA5 data as in Belmonte Rivas and Stoffelen (2019) for example. Reanalyses fields are affected by the observations being assimilated, using spatial structure functions, which are posed as stream function and velocity potential "blobs", defined based on forecast ensemble statistics. These increments may not treat vorticity fields well and produce noise. Another reason may be in interpolation of the vorticity fields, but where no details are provided.

314: All steps appear rather ad hoc, but together they define a vortex isolation and data procedure. Moreover, it appears as a community procedure, as others elaborated similar procedures. Does the procedure work similarly well for other reanalyses, mesoscale models or the operational ECMWF analysis? To me, it appears tuned to the characteristics of your input ERA5 fields. Perhaps mention that other meteorological model fields may require further tuning of the vortex detection procedure.

316: established -> constructed

318: Terrain-induced flows are normally tied to the terrain and not to the wind, hence presumably they'd typically not produce vortex tracks according to your criteria?

320: established -> comparison ; recall that AVHRR are not a direct measurement of PMC, cf. comment 113.

455: The concept of environmental wind speed is not clear. What is its use? The 10m wind vector around a moving vortex is rather variable, depending on steering flow and vortex strengths. The baroclinic nature of these high-latitude vortices makes their surface appearance usually asymmetrical. I can understand you'd like to capture this, but this is not clear from the text. Please clarify what relevant dynamical characteristics can be extracted. Fig. 11b appears a vortex interacting with land and hence surface winds are distorted?

458: To first order, the destructive force goes with the third power of the wind speed, irrespective of it is generated by the environmental flow, vortex contribution or related to local convection, all count. In open sea, the waves, build by the wind, are of course very important as well, as the dimensions of the structures at sea may resonate with long and forceful waves.

470: The scatterometer section is rather poor as scatterometers, in particular ASCAT, reveal detailed dynamical PL characteristics. Wind vectors fields reveal the exact surface position, structure and divergence and curl and with high coverage. See also comment 208. Unfortunately, not much has been published on active satellite surface winds and PLs, while Furevik et al. (2015) provide some overview.

476: "measurable wind patterns"? My experience in scatterometry for PL/MPC is that tracking is very well feasible and measurable. I copy below a slide I show in nowcasting training using https://scatterometer.knmi.nl/tile_prod/index.php . For a description, see the figure above. Several things to note here: 1) Many scatterometer acquisitions exist over a day to verify both model dynamics (green arrows) and IR images (grey-scale). 2) IR clouds follow the dynamics seen at the surface, i.e., the dynamics produce clouds in upward motion and dissolve clouds in downward motion, i.e., the clouds follow the winds. 3) Initially, a through appears in the scatterometer winds below a cloud shield, where the green arrows are not informed by it initially. As scatterometer winds are assimilated at ECMWF the disturbance appears in the model data over the day. As mentioned earlier, L3 and L4 products are produced with scatterometer information, model information, incl. ERA5, and fields of spatial derivatives. These appear more ideal to "measure" model and, after collocation, AVHRR characteristics in PL/MPC than the rather unfavorable diagnostics presented here.

[Figure]

22 hours
at 65S
on 19 May
2023

475: You find many tracks that are not in AVHRR. Following the comment above, this could well be because the vortical structure is not well expressed in the clouds. Observed dynamics at the surface may prove a better way to verify these vortices. A problem here is that scatterometer winds are only consulted after a imperfect AVHRR filter, rather than before this filter. This can be done by exploiting collocated model and scatterometer data and their spatial gradients, which are available. When only one scatterometer is available (up to 2007), then track cannot be well verified, but every occasion a vortex appears in a scatterometer swath verification may be done. That would results in hits and misses of ERA5 vortices, which verify your product more substantially in my view.

486: 3 hours implies three points, right? 50% in these cases implies only 2 of 3 points and 80% 3 of 3 points and 4 of 5 hits for longer tracks for example. It is clear that adding more lenient vortex criteria will improve apparent skill as the Stoll data set is fixed. It does not necessarily imply better performance though as Stoll. How much false tracks/points do you add?

493: demonstrates? Clearly, wind variability is high in cold air outbreaks near the surface and upper air interaction more fierce. Allowing more noise in ERA5 vorticity or more lenient

vortex criteria will reveal more tracks, but are they reliable? If some of them appear in the proximity of observed tracks, it appears insufficient to demonstrate capability. How many unverified tracks are produced (false alarms)? Could these accidentally be added to the hit list? In that case skill is not enhanced, but rather PL/MPC noise is added.

495: So, ERA5 finds about 10 times more PLs/MPCs than the most extensive observational data set (Rojo). Is this noise? Looking at your AVHRR score, noise appears indeed manifest; 57,688 ERA5 vortex tracks, only 1,184 or 1 in 50 are confirmed. This may be related to the fact that AVHRR is a rather indirect measure of vortical activity, while you appear to appreciate the skills of AVHRR. What are the >90% misses in your data set? As these amounts appear rather overwhelming, it appears very relevant to understand their characteristics if these are used for geophysical analyses or trend analyses. The difference with Stoll's 3179 tracks from the same ERA5 is also rather overwhelming. What are the differences? I further understand less than 700 (only about 1%) remain for further comparison. I'm concerned what the other 99% represent?

507: demonstrate that such discrepancies are not errors -> characterize such discrepancies ; they are errors as ERA5 uniquely represents PLs/MPCs.

509: stable -> negligible

512: remove "stable"; the choice of this word is a bit concerning, does it imply that you favor a smooth representation of disturbances? Spatial smoothing is applied, but it can obviously kill PLs/MPCs, which is a negative effect. If Stoll uses data from ERA5 that is less interpolated, it may in fact be a good thing that it represents more variability? Please elaborate in your manuscript.

532: Please indicate in the figure legend what percentage of the most favorable (matched) cases it represent. The non-matched cases are less detectable and probably have much less favorable verification.

541: "Despite" or "Due to"? Less favorable cases may not match well?

544: "reasonable"; you allow a 120 km separation and then one gets separations with a SDD of about 120 km, which implies little skill. Do you reason for little skill? Presumably, further work is needed to explain the lack of skill? Better explain to the users what further work would be appropriate in this discussion.

559: How do you know what these cases are? They have not been verified, at least not in the manuscript. Could they not be numerical artefacts? Are they associated with real features or are these ERA5 simulated features?

565: Is the point not how reliable ERA5 is to represent PLs and PMCs? One could test that using the cases where verification is available and determine and not yet used in ERA5 (by data assimilation). Therefore, testing ERA5 background states, winds are independent of any new observations, one could establish the capability of ERA5 to predict PLs and MPCs. Only after this, ERA5 can be used with confidence for associated geophysical studies in my view. Would you agree?

580: As explained above, observations directly associated with PL/MPC dynamics may be further exploited to characterize these systems and the fidelity of reanalyses to represent them.

**Additional references:**

Furevik, B. R., H. Schyberg, G. Noer, F. Tveter, and J. Röhrs, 2015: ASAR and ASCAT in Polar Low Situations. *J. Atmos. Oceanic Technol.*, **32**, 783–792, https://doi.org/10.1175/JTECH-D-14-00154.1.

King, Gregory P., Marcos Portabella, Wenming Lin, and Ad Stoffelen. 2022. "Correlating Extremes in Wind Divergence with Extremes in Rain over the Tropical Atlantic" Remote Sensing 14, no. 5: 1147. https://doi.org/10.3390/rs14051147 .

Skamarock, W. C., 2004: Evaluating Mesoscale NWP Models Using Kinetic Energy Spectra. Mon. Wea. Rev., 132, 3019–3032, https://doi.org/10.1175/MWR2830.1 .

Vogelzang, Jur, and Ad Stoffelen. 2022. On the Accuracy and Consistency of Quintuple Collocation Analysis of In Situ, Scatterometer, and NWP Winds, *Remote Sensing* 14 (18), 4552, https://doi.org/10.3390/rs14184552

---

## Author Comment (AC1)

The authors are grateful to the editor and all reviewers for their time and energy in providing helpful comments that have improved the manuscript. In our revised paper, we further re-checked all revisions and performed grammatical corrections to help readers understand our manuscript easier.

In this document, reviewer' comments have been addressed point by point. Referee comments are shown in black italics and author responses are shown in blue regular text and revised version of the manuscript is shown in green text.

**Reviewer #1:**

**Major comments:**

The manuscript presents the Integrated Multi-source Polar Meso-Cyclone Tracks (IMPMCT) dataset based on both ERA5 reanalysis and remote sensing data during winter in the Nordic Sea, demonstrates clearly the workflow of this method, and compares the results with existing manually identified and reanalysis-based track datasets. There remains a clear need for establishing a more comprehensive tracking dataset capable of capturing PMCs throughout their lifecycle due to their impacts on human activities and regional climate change. The manuscript is generally well-organized, and the figures effectively communicate the results while being concise. However, there are a few aspects where the presentation could be improved. The detailed comments are listed below, and I encourage the authors to make the necessary adjustments to improve the study.

1. The present study utilized a series of datasets, including ERA5 reanalysis, AVHRR data andQuikSCAT/ASCAT data, which have different spatial and temporal resolutions, and these data are stored with different projections/grids. How are these multi-source datasets treated in the cyclone tracking algorithm to maintain consistency? Please clarify.

**Re:** Thank you very much for your inquiry. We fully understand your concern about the data matching method and the accuracy of the tracking algorithm. Issues such as spatial-temporal resolution and potential representativeness errors are indeed key issues that must be handled carefully in dataset establishment. We provide the following detailed explanations:

(1) data matching

- **ERA5-AVHRR Matching**: Vortex centers from ERA5 (hourly vorticity fields) were matched to AVHRR cloud features within a 1-hour window and 250-km radius. AVHRR data validated genuine cyclone evolution. Trajectories were excluded if AVHRR temporal resolution was insufficient to confirm evolution or if the average matching distance exceeded 150 km.
- **QuikSCAT/ASCAT**: Wind data supplemented cyclone attributes but did not drive identification. Matches to AVHRR were constrained to a 30-minute window. Scan timestamps are provided for error assessment.

(2) data grids:

We used a VCI(Vortex-Centered Infrared) grid, which is a conformal projection grid. This grid has mutually perpendicular meridians and parallels, with shape invariance under translation, and local equidistant characteristics. It preserves local isotropy and enables consistent spatial calculations (see **[Line 353-363]** for details).

2. Line 164: ERA5 data. How accurate are the ERA-5 fields used in the analysis of the Nordic Sea? What are the known biases? As the authors did not repeat their method with other reanalysis datasets to test the robustness of their results, I would suggest

declaring the known biases of ERA5 in this part.

**Re:** Your reminder is very important, which helps to improve the rigor of the study. We have added descriptions about the quality of ERA5 regarding meteorological elements related to polar mesocyclones in the Nordic Sea in the revised version, see **Line 169-179**.

This additional ERA5 data description is now described in the revised version of the manuscript:

ERA5 reanalysis dataset demonstrates robust performance in representing meteorological fields over the Nordic Seas, such as sea level pressure, air temperature, and humidity (Graham et al., 2019; Moreno-Ibáñez et al., 2023; Yao et al., 2021). Most notably, its effective characterization of cold air outbreaks has been proven to correlate closely with the timing and location of PLs (Meyer et al., 2021). However, beyond the previously mentioned underestimation of near-surface strong winds in Section 1, Wang et al. (2019) found ERA5 data exhibits a warm bias over Arctic sea ice during winter and spring, which makes it difficult to accurately simulate the frequently occurring strongly stable boundary layers prevalent in winter and early spring. Consequently, the intensity of PMCs near the sea ice edge might be overestimated. Nevertheless, more accurate total precipitation and snowfall data in ERA5 (Wang et al., 2019) significantly benefits the representation of enhanced latent heat release mechanisms associated with PLs (Moreno-Ibáñez et al., 2021).

3. Line 262: To maximize the inclusion of potential PMCs, we implement more lenient vortex detection criteria compared to Stoll et al. (2021). The selected criteria seem to be very subjective. Importantly, how sensitive are the results to subjective criteria such as the "vorticity peak threshold", "isolated vortex threshold"? Have the authors conducted sensitivity tests, and what metrics were used to evaluate the robustness of the results? Please include this.

**Re:** Your suggestion is very important. Following your advice, we have deleted the statement that directly adopts lenient criteria to avoid confusing readers. Meanwhile, we have added a subsection "3.1.3 Sensitivity experiments of vortex identification parameters", in which we supplemented two groups of sensitivity experiments on vortex identification parameters. We also calculated the matching rates of vortex tracks obtained from different parameter sets with other PL lists.

Through the experiments, we found that:
- Lowering the vorticity peak threshold ($\zeta_{max0}$) increased detection of weak vortices (lifespan +3 hrs) and nearly doubled capture of moderately weak systems. **[Line 316-322]**
- Reducing the isolation threshold ($\gamma$) improved sensitivity to splitting events but shortened mean vortex lifespan by ~2 hrs due to increased transient sub-vortices. **[Line 323-328]**
- Experiment a was chosen to maximize weak-PMC inclusion and validation against PL datasets (Table S2) shows the lenient-threshold vortex tracks consistently yield higher matching rate. **[Line 527-532]**

This additional **Sensitivity experiments** is now described in the revised version of the manuscript:

To evaluate the sensitivity of vortex identification parameters, we conducted three sensitivity experiments with the following configurations, each designed to test the impact of varying key thresholds $\zeta_{max0}$ ($\zeta_{min0}$) and $\gamma$ on vortex detection:
1) Experiment a (lenient thresholds): $\zeta_{max0} = 1.2\times10^{-4}$ s$^{-1}$, $\zeta_{min0} = 1.0\times10^{-4}$ s$^{-1}$, $\gamma = 0.15$;

2) Experiment b (intermediate thresholds): $\zeta_{max0}$ = 1.2×10$^{-4}$ s$^{-1}$, $\zeta_{min0}$ = 1.0×10$^{-4}$ s$^{-1}$, $\gamma$ = 0.25;

3) Experiment c (strict thresholds, following Stoll et al. 2021): $\zeta_{max0}$= 1.5×10$^{-4}$ s$^{-1}$, $\zeta_{min0}$ = 1.2×10$^{-4}$ s$^{-1}$, $\gamma$ = 0.25

The influence of threshold variations on vortex detection characteristics was systematically evaluated by analyzing differences in the number of identified vortex tracks, their lifespans, and their vorticity across the three experiments. As shown in Fig. 7, threshold adjustments predominantly affected vortices exhibiting maximum vorticity ($\zeta_{\text{trmax}}$) less than 2×10$^{-4}$ s$^{-1}$, with distinct impacts observed for changes in $\zeta_{max0}$ versus $\gamma$. The principal findings are:

First, focusing on the impact of $\zeta_{max0}$ (by comparing Experiment b, which uses a lenient $\zeta_{max0}$, with Experiment c, which uses a strict $\zeta_{max0}$), we found that the lenient threshold in Experiment b captured an additional 8,077 weak-vorticity tracks (with $\zeta_{\text{trmax}}$ < 1.5×10$^{-4}$ s$^{-1}$). This adjustment also extended the mean lifespan of detected vortices by approximately 3 hours. Under the 6-hour minimum lifespan criterion—used to filter transient disturbances—this extension nearly doubled the detection rate of moderately weak vortices (1.5×10$^{-4}$ s$^{-1}$ < $\zeta_{\text{trmax}}$ < 2×10$^{-4}$ s$^{-1}$), highlighting the importance of $\zeta_{max0}$ in capturing less intense but persistent systems.

Second, examining the role of $\gamma$ (by comparing Experiment a, which uses a lenient $\gamma$, with Experiment b, which uses an intermediate $\gamma$) revealed that the lenient $\gamma$ threshold in Experiment a increased the count of weak-to-moderate vortices (1.5×10$^{-4}$ s$^{-1}$ < $\zeta_{\text{trmax}}$ < 3×10$^{-4}$ s$^{-1}$). This increase was attributed to enhanced sensitivity to vortex splitting events, though it came with a trade-off: the mean lifespan of detected vortices was reduced by approximately 2 hours, likely due to more frequent identification of short-lived sub-vortices during splitting

Given the objective of constructing a comprehensive dataset capturing the full spectrum of PMCs, including weaker systems potentially omitted by stricter criteria, the parameter set from Experiment a was ultimately selected. This configuration yielded the highest number of vortex tracks, thereby ensuring the inclusion of marginally intense or transient PMCs and providing a more robust foundation for subsequent analysis. Validation of these results against established polar low datasets is presented in Section 4.

[Figure]

Figure 1 Sensitivity analysis of vortex identification parameters across different maximum track vorticity groups: (a) number of identified tracks, (b) mean track lifetime.

This **matching rate of the reanalysis-based track dataset with different vortex identification parameters compared to other PL track datasets** is now described in the revised version of supplement Table S2:

| Experiment | Track counts | Matching rate(%) with | | |
|---|---|---|---|---|
| | | Stoll | Rojo | Noer |
| a | 59975 | 93.68 | 69.73 | 87.72 |
| b | 52708 | 92.04 | 68.11 | 86.84 |
| c | 33622 | 87.39 | 61.35 | 80.70 |

4. It seems a YOLO (You Only Look Once) object detection algorithm is employed to detect and extract cyclonic cloud characteristics. This description of this procedure could be improved in my opinion. The authors start by generally describing the structure of the YOLOv8-obb model on line 377, with so many acronyms. However, the specific process by which this algorithm works to detect cloud features was oversimplified in the following paragraph.

Re: Thank you for your comment. We simplified the YOLOv8-obb-pose description by removing technical acronyms (e.g., decoupled head module) and retained only the framework overview. Algorithm details are deemphasized as YOLO is established.

Meanwhile, we have supplemented detailed examples of the algorithm's recognition results to help readers understand and reproduce the relevant recognition process more easily, as shown in Figure 10. **[Line 412-416]:**

The network architecture of the YOLOv8-obb-pose model comprises three main components: Backbone for multi-dimensional feature extraction, Neck for enabling multiscale feature fusion, and Head for extracting cyclone type, center coordinates, and oriented bounding box parameters (e.g., length, orientation). As shown in Fig. 10, the YOLOv8-obb-pose model successfully detects two spiral clouds (Fig. 10a) and two comma-shaped clouds (Fig. 10b) in VCI images, with oriented bounding boxes,cyclone type and center points marked.

**(a) detected spiral clouds**      **(b) detected comma clouds**

[Figure]

[Figure]

Figure 2: Examples of cyclonic cloud detection using the YOLOv8-obb-pose model: (a) two spiral clouds detected in a VCI image and (b) two comma-shaped clouds detected in a VCI image. The oriented bounding boxes for spiral clouds are shown in purple, and for comma-shaped clouds in blue. The centers of the cyclones are marked with green points. The cyclone type and detection confidence are displayed above each bounding box.

Additionally, a description of how to process the detection results to extract cyclone information is added. This helps clarify the role of the YOLOv8-obb model within the overall algorithmic workflow. **[Line 427-431]:**

For each detected cyclone, the center coordinates and the four vertices of the oriented bounding box are converted back to geodetic coordinates using the inverse of Eq. (1) and (2). The lengths of the four sides of the bounding box are calculated using the haversine formula, with the cyclone's length (width) defined as the mean size of the two long (short) sides of the rectangle. The geographic coordinates of the cyclone center are then used for subsequent matching with vortex centers.

5. When comparing the results from the IMPMCT to existing identified PL lists from previous studies, the authors give the difference in parameters and plot them. It is more appropriate to conduct a significance test between two samples in order to statistically validate the accuracy.

 **Re:** Your opinion is very important. In comparing with other datasets, in addition to parameter difference indicators, consistency is also an important verification goal. Since there is no absolutely accurate true value dataset, we adopted the Bland-Altman analysis method for comparison. This method provides an intuitive and easy-to-understand way to evaluate the consistency of measurement values of the same object

obtained by different technical means. If the distribution of differences between the two measurement results is normal, 95% of the differences should be within ±1.96 times the standard deviation of the differences, and we call this interval the 95% limits of agreement. This method evaluates the degree of agreement between the two methods by quantifying the mean difference (bias) and limits of agreement (LoA). If the vast majority of differences fall within the limits of agreement, it can be considered that the two methods have good consistency.

Results show:

- 95% of differences in vortex/cyclone properties fall within ±1.96 SD of the mean difference (Sec. 4; Table 3; Fig. S1).
- Small biases exist (e.g., mean difference: −6.8 km in vortex diameter; 0.3 hPa in SLP), attributable to methodology differences. **[Line 594-599]**.

This additional consistency test is now described in the revised version of the manuscript:

To statistically validate the agreement between IMPMCT and the Stoll (2022) dataset and Rojo list regarding vortex and cyclone properties, we performed Bland-Altman analysis (Bland and Altman, 1999). This method assesses the agreement between two measurement techniques by quantifying the mean difference (bias) and the limits of agreement (LoA), defined as the mean difference ± 1.96 standard deviations of the differences. A summary of the Bland-Altman analysis for key properties is presented in Table 3. The corresponding Bland-Altman plots, illustrating the distribution of differences against the average values for each property, are provided in Supplementary Fig. S1. As shown in Table 3, the vortex properties derived from ERA5 reanalysis data exhibit a slight systematic bias compared to other datasets. This bias is likely attributable to differences in computational methods. Critically, the Bland-Altman analysis confirms strong agreement, with approximately 95% of the differences for each property falling within the respective 95% limits of agreement (Table 3, last column), supporting the consistency between the datasets.

**Table 1 Property difference between IMPMCT and other PLs list**

| Property | Matched number | Mean Difference | Standard Deviation of Differences | % Points within LoA |
|---|---|---|---|---|
| 850 hPa relative vorticity ($10^{-5}$ s$^{-1}$) | 21281 | 0.61 | 2.15 | 95.1 |
| SLP (hPa) | 14522 | 0.3 | 0.76 | 95.7 |
| vortex equivalent diameter (km) | 21281 | -6.8 | 39.46 | 93.7 |
| track-max near-surface wind speed (m s$^{-1}$) | 42 | -0.27 | 4.83 | 95.2 |
| cyclone cloud diameter (km) | 892 | 6.76 | 121 | 94.7 |

This additional consistency test plotting is now described in the revised version of the supplyment:

[Figure]

Figure S3 Bland-Altman analysis of Property Differences Between IMPMCT and Other PL list.(a) 850 hPa relative vorticity, (b) vortex equivalent diameter, (c) SLP, (d) cyclone cloud diameter.The x-axis represents the mean property value of IMPMCT and the other dataset; the y-axis represents the difference in properties (IMPMCT minus PL list). Point color indicates Gaussian kernel density. The black dashed line denotes the zero line. The red solid line indicates the mean difference of the sample properties. The upper and lower green dashed boundaries represent the limits of agreement (LoA), defined as the mean difference ± 1.96 standard deviations of the differences.*Note: Differences for properties (a), (b), and (c) are comparisons between IMPMCT and the Stoll (2022) dataset, whereas (d) uses the Rojo list. The difference analysis for track-max near-surface wind speed is not shown due to insufficient sample size.

6. Figure issues
- Specify what is plotted in Figure 1 in the name of the colorbar, same comments for Figure 3b, and Figure 7.
- The green star symbols denoting the local vorticity maxima are hard to read when overlaid on the AVHRR infrared imagery. Please change the color or enlarge the symbols. Same comments for stars in Figure 10b and wind vectors in Figure 11.
- The unit of the colobar in Figure 7a should be $1e^{-4}s^{-1}$

**Re:**
- Colorbar labels added to Figs. 1, 3b, 8.
- Symbols were enlarged for visibility (Figs. 1, 8, 10b, 12).
- Unit corrected in Fig. 8a to $10^{-4}s^{-1}$.

These figures have been modifiedied in the revised version of the manuscript:

[Figure]

Figure 4: Two AVHRR satellite images. (a) A PMC in Barents Sea. (b) A PL in Norwegian Sea. The yellow stars mark the centers of these two cyclones.

[Figure]

Figure 5: (a) 850 hPa relative vorticity field obtained by ERA5 data. (b) AVHRR infrared imagery concurrent with the time step in (a). The shading represents 850 hPa relative vorticity smoothed over a uniform 60 km radius and local vorticity maxima are marked by green star symbols, while regions enclosed by solid black contours denote the unpartitioned-vortex zone.

[Figure]

Figure 6: Two examples of VCI image generation. For the two vortices shown in (a), the AVHRR IR image (b) reveals a polar low located to the east of vortex 1 and vortex 2. This polar low exists simultaneously in the VCI images centered on vortex 1 and vortex 2 (c, d). The shading in (a) represents 850 hPa relative vorticity smoothed over a uniform 60 km radius, with gray contour lines indicating sea-level pressure at 10 hPa intervals. The centers of vortex 1, vortex 2, and the polar low are respectively marked by green, red, and yellow stars.

[Figure]

Figure 7: (a) A matched vortex track and cyclone track and (b) partial corresponding VCI images. For (a), blue solid line represents the vortex track at hourly resolution, while grey solid line with green points depicts the cyclone track points formed in VCI images that correspond one-to-one with vortex points. The color of the track points indicates the magnitude of relative

vorticity at each vortex point. For (b), the cyclone develops sequentially from left to right and top to bottom, with scan intervals between images approximately six hours apart.

[Figure]

Figure 8: VCI images overlaid with near-surface wind speeds for cyclones exhibiting strong (a) and weak (b) local impacts on near-surface wind conditions. Color shading represents QuickSCAT-measured 10m near-surface wind speeds, with green arrows indicating corresponding wind vectors. Yellow borders denote the cyclones' bounding oriented box. Blue and red circular borders respectively represent the short and long search ranges. Yellow and red stars indicate the cyclone center and maximum wind speed point locations.

**Minor comments:**

1. Lines 41-42: Add references about this statement.

**Re:** We have supplemented two relevant references and revised some expressions at **[Line 41-43]**:

"Polar Mesoscale Cyclones (PMCs) are mesoscale cyclonic weather systems that frequently occur over open waters or sea-ice edges in regions poleward of the main polar front zones (Condron et al., 2006; Rasmussen and Turner, 2003)."

2. Lines 59-61: Add references about this statement or remove it as it seems irrelevant to the core points of this paragraph.

**Re:** We have removed the initial broad statement about the effectiveness of remote sensing. Starting directly with the core distinction criteria better aligns with the paragraph's main purpose: "Cyclonic cloud morphology and surface wind fields serve as the primary criteria..." **[Line 59-60]**.

3. Lines 129-131: Moreover, fundamental questions persist regarding the differences in formation mechanisms between PMCs and PLs, and whether PMCs can transition into PLs under specific meteorological conditions. This question seems not to be addressed.

**Re:** The speculative sentence on PMC-PL transition mechanisms was deleted. **[Line 127]**

4. Line 138: Winter should be defined here rather than in the Data part.

Re: The seasonal coverage of the data has been added to both the Abstract **[Line 23]** and Introduction **[Line 135].**

5. Line 140: "multi-dimensional" to "multiple"

Re: Done**. [Line 144]**

6. Line 161: "sourced" to "obtained"

Re: Done**. [Line 158].**

7. Line 169: delete "for atmospheric, land, and ocean variables"

Re: Done**. [Line 166].**

8. Lines 191- 192: Notably, QuikSCAT data spans only 1999–2009, while ASCAT has remained operational since 2010. Rephrase to: QuikSCAT operated from 1999 to 2009, whereas ASCAT has continued operations since 2010.

Re: Done**. [Line 200].**

9. Lines 281-284: "Specifically, for a vortex at a given time step, its ideal point after experiencing a time step under the steering wind influence is first calculated A search radius of 180 km is then applied around this estimated location to facilitate vortex tracking in subsequent time steps.." Should be two separate sentences.

Re: Done. **[Line 281]**

10. Lines 293-294: Rephrase to: If no spatially connectable vortices are identified in adjacent time steps, the vortex is classified as being terminated.

Re: Done. **[Line 291-292]**

Lines 316-319: Rephrase to: Building upon the lenient vorticity identification criteria established in prior analysis, a substantial population of vortex tracks has been identified within the reanalysis dataset. This collection encompasses not only cyclonic systems but also terrain-induced shear flows, low-pressure troughs, and small-scale atmospheric disturbances.

Re: Done. **[Line 349-352]**

12. Line 373: Delete "deliberately"

Re: Done. **[Line 404]**

11. Lines 391-393: Rephrase to: To ensure prediction stability, particular emphasis is placed on maintaining consistent oriented bounding box annotations and center point positions across similar evolutionary phases of cyclonic cloud morphologies.

Re: Done. **[Line 419-421]**

12. Linee 409-413: Rephrase to: To remove duplicate records, we implement a selection criterion: for any cluster of detections from the same AVHRR infrared scan (with cyclone centers <50 km apart), only the detection whose center is nearest to the VCI image center is retained.

**Re:** Done. **[Line 445-448]**

13. Lines 453-455: Rephrase to: To reduce the influence of strong winds in the cyclone core, we use the 75th percentile of wind speeds within the extended search radius as the environmental advection speed (reference value).

**Re:** Done. **[Line 488-490]**

14. Lines 484-485: Rephrase to: All reference datasets are spatially and temporally co-located with our derived tracks, retaining only those persisting for ≥3 hours.

**Re:** Done**. [Line 517-518]**

15. Line 526: "extraneous" to "irrelevant"

**Re:** Done. **[Line 563]**

16. Line 545: Rephrase to: Additionally, since the dataset includes remote sensing images of cyclones, users can easily verify the accuracy of cyclone properties and make necessary adjustments based on their specific use cases.

**Re:** Done. **[Line 581-582]**

17. Line 568: "these categories" to "them"

**Re:** Done. **[Line 614]**

**References**

Belmonte Rivas, M. and Stoffelen, A.: Characterizing ERA-interim and ERA5 surface wind biases using ASCAT, Ocean Sci., 15, 831–852, https://doi.org/10.5194/os-15-831-2019, 2019.

Graham, R. M., Hudson, S. R., and Maturilli, M.: Improved performance of ERA5 in arctic gateway relative to four global atmospheric reanalyses, Geophys. Res. Lett. , 46, 6138–6147, https://doi.org/10.1029/2019gl082781, 2019.

Han, Y., & Ullrich, P. A. The system for classification of low-pressure systems (SyCLoPS): An all-in-one objective framework for large-scale data sets. JGR Atmospheres, 130, e2024JD041287. https://doi.org/10.1029/2024JD041287,2025.

Moreno-Ibáñez, M., Laprise, R., and Gachon, P.: Recent advances in polar low research: current knowledge, challenges and future perspectives, Tellus: Series A, 73, 1–31, https://doi.org/10.1080/16000870.2021.1890412, 2021.

Wang, C., Graham, R. M., Wang, K., Gerland, S., and Granskog, M. A.: Comparison of ERA5 and ERA-interim near-surface air temperature, snowfall and precipitation over arctic sea ice: Effects on sea ice thermodynamics and evolution, The Cryosphere, 13, 1661–1679, https://doi.org/10.5194/tc-13-1661-2019, 2019

---

## Author Comment (AC2)

The authors are grateful to the editor and all reviewers for their time and energy in providing helpful comments that have improved the manuscript. In our revised paper, we rechecked all revisions and performed grammatical corrections to help readers understand our manuscript more easily.

This document addresses reviewer comments point-by-point. Reviewer comments are presented in black italics, author responses in blue regular text, and revised manuscript text in green text.

**Reviewer #2:**

**General comments:**

This manuscript describes a polar mesoscale cyclone dataset assembled by the authors for the study of polar meteorology. In general, it is well-written, and the steps in dataset construction are well-described. The dataset compiled here definitely is useful for the community. I have some minor suggestions and comments. Once these issues are addressed, I recommend publishing this manuscript.

Re: We sincerely appreciate your assessment of the dataset and will rigorously consider all recommendations.

**Minor comments**

1. Title: I don't see a reason to use "Meso-Cyclone" as mesocyclone is a well-defined term in meteorology. It is just one word. Also, it is a database only for the Nordic Seas, not the entire polar oceans. It would be better to reflect this regional context in the title, or in the acronym defined for the dataset.

Re: Thank you for this important observation. We have revised the title to:

" IMPMCT: a dataset of Integrated Multi-source Polar Mesoscale Cyclone Tracks in the Nordic Seas."

2. Line 200, "For each vortex with available AVHRR data, …" What percentage of such vortices identified from ERA5 have AVHRR data available? This info is useful for readers. As AVHRR is on a sun-synchronous satellite, it does not have full synoptic coverage for the polar region. So the percentage of actual coverage in this context needs to be described.

Re: We thank you for highlighting this essential aspect for reproducibility. The

Results section now includes:

After excluding vortex tracks with >60% land presence (~20% reduction), 47,167 tracks remained for AVHRR matching. Matching required: (1) full 200-km radius coverage for individual points, (2) ≥2 matched points within ±3h of peak vorticity and ≥6 points per track lifetime. Fig. 13 shows wintertime (Nov-Apr) matching statistics: 43% of points and 61% of tracks matched on average. Only ~3% of matched tracks were incorporated into the IMPMCT dataset. This low inclusion rate stems from cloud obstruction, cloud-ice contrast limitations, temporal resolution constraints, and detection methodology (e.g., higher 2001 inclusion reflects meticulous manual identification, while 2023's lower rate resulted from post-publication incidental discoveries). Crucially, IMPMCT's cyclone proportion underestimates true PMC prevalence, as many low-cloud PMCs lack discernible features.

[Figure]

Figure 13: Annual winter (November-April) time series: (a) ERA5-derived vortex points (green), available AVHRR files (red), and AVHRR-matched vortex points (blue). (b) Ratio of AVHRR-matched vortex tracks to ERA5-derived tracks (yellow), and ratio of IMPMCT tracks to AVHRR-match tracks (purple). Note: Bars represent distinct categories (not stacked)

3. It is not clear to me what exact cloud properties are included in the dataset beyond the cloud morphology. Any usual cloud properties such as cloud top pressure, cloud optical depth, etc., are included? If so, it's better to specify them up front.

**Re**: Thank you for highlighting this need for clarity. The Abstract has been revised as follows:

The dataset contains 1,172 vortex tracks, 16,561 cyclonic cloud features (length, width, morphological characteristics (spiral/comma shape, center position), and 4,588 wind speed records (wind vector imagery and cyclone maximum winds). Corresponding ERA5-derived hourly vortex tracks are also provided, including 850-hPa vorticity and proximate sea-level pressure minima.

4. Table 2: Why is the matched fraction with Rojo PL tracks so low compared to the matches with the other two PL track datasets? This needs to be explained

**Re**: We sincerely appreciate your attention to this detail. Our analysis reveals two key factors for Rojo's lower match rate (71% vs. Noer's 85%): First, Rojo's direct AVHRR identification contrasts with Noer's model-interpolated hourly centers, creating greater ERA5 deviation. Second, Rojo includes secondary PL centers (54% match rate) that ERA5 resolves poorly versus major centers (80%), consistent with Stoll (2022). Fig. 1 exemplifies a frequent mismatch case where ERA5's nearest vortex center was 227 km from Rojo's observed position. For Stoll's data, we introduced "vortex matching" (99% match) to address vorticity peak misalignments from smoothing differences (Fig. S2). The manuscript text now explains:

To further investigate mismatches between the reanalysis-based tracks and existing PL datasets, we implemented a nearest-point matching analysis (Table 2). A successful nearest-point match was recorded when a PL center from any list had at least one co-temporal vortex center within 120 km (60 km for the Stoll dataset). The track-level mismatches primarily stemmed from these point-level discrepancies. Crucially, the methodological differences between datasets explain the variation: While the Noer list derives from numerically modeled and AVHRR-assimilated hourly positions (typical of operational forecasting systems), the

Rojo list relies on direct AVHRR identification at irregular temporal intervals, resulting in greater deviation from ERA5 representations. Furthermore, the Rojo compilation includes numerous secondary PL centers—features inherently less resolved by reanalysis data (Stoll, 2022)—whereas Noer focuses primarily on dominant PLs of operational significance. This distinction is clearly reflected in our analysis: Major PL centers (n=2,527) exhibited an 80% matching rate, while secondary centers (n=1,115) showed significantly lower alignment (54%), thereby reducing Rojo's overall match rate. For the Stoll dataset, we additionally calculated a vortex matching rate (Table 2), counting a match when a Stoll center fell within the spatial domain of its nearest co-temporal vortex. This metric primarily addresses positional offsets caused by vorticity peak misalignment, which appears attributable to differences in smoothing algorithms (illustrated in Fig. S2). Our implementation seems to employ stronger uniform smoothing compared to Stoll's methodology, explaining why more lenient identification thresholds yield superior track matching with Stoll's dataset.

[Figure]

**Figure 1: (a) 850 hPa relative vorticity field obtained by ERA5 data. (b) AVHRR infrared imagery concurrent with the time step in (a). The shading represents 850 hPa relative vorticity smoothed over a uniform 60 km radius and local vorticity maxima are identified by green star symbols, while regions enclosed by solid black contours denote their borders.The red star symbol marks a mismatched cyclone center from Rojo's PLs list, while the black star symbol marks the nearest local vorticity maxima from the cyclone center (227 km).**

**Table 1: the matching rate of the reanalysis-based track dataset for IMPMCT generation compared to other PL track datasets.**

| PL tracks | Time period | Tracks in Nordic Sea (>3hr) | Track matched fraction(%) | Points | Nearest points matched fraction(%) | Vortex matched fraction(%) |
|---|---|---|---|---|---|---|
| Noer | 2002-2011 | 114 | 87 | 1670 | 85 | - |

| | | | | | | |
|---|---|---|---|---|---|---|
| Rojo | 2000-2019 | 370 | 69 | 3642 | 71 | - |
| Stoll | 2000-2020 | 3179 | 93.68 | 75650 | 93 | 99 |

[Figure]

**Figure S2: ERA5 850-hPa fields: (a) Relative vorticity. (b) Uniform 60-km smoothed vorticity. Vorticity field comparison showing center displacement between Stoll (blue points) and our detection (green points)**

5. Line 598 "IMPMCT could serves as a critical benchmark for evaluating high-latitude climate model performance." It would be beneficial to elaborate on how a track-based dataset can be utilized for climate model evaluation. Are the tracks compiled here enough for robust statistics (related to comment #2 above)? What standard model output can be used directly for such comparison, or do climate models need to output high-resolution data to be used by track algorithms to generate similar datasets for comparison?

**Re**: We apologize for the oversight. A more precise statement would reference "numerical weather prediction models". The revised text clarifies:

The IMPMCT dataset serves as a critical benchmark for evaluating high-latitude numerical weather prediction model performance, while simultaneously functioning as a unique case library for comparative studies of polar lows (PLs) and polar mesoscale cyclones (PMCs) concerning their formation mechanisms, intensity thresholds, and sea-ice interaction dynamics. Furthermore, it constitutes an essential resource for enhancing polar maritime hazard forecasting. The repository of cyclone cloud morphology facilitates automated identification of model-undetected systems. This is enabled by advanced deep learning frameworks, enabling systematic evaluation of model representation fidelity for PLs/PMCs. From a climatological perspective, this resource permits establishment of comprehensive

objective identification criteria based on reanalysis data, thereby enabling robust analysis of climate-scale trends and genesis potential shifts in PL/PMC activity (Stoll, 2022; Zhang et al., 2023).

6. There are occasional English typos, e.g., "could serve" not "could serves" at Line 598. A careful proofread would be helpful. I assume ESSD might have a technical editor in a later stage for such proofreading.

**Re**: We thank you for this observation. Comprehensive grammatical and spelling checks will be implemented throughout the revised manuscript.

**Reference**

Stoll, P. J.: A global climatology of polar lows investigated for local differences and wind-shear environments, Weather Clim. Dynam., 3, 483–504, https://doi.org/10.5194/wcd-3-483-2022, 2022.

Zhang, X., Tang, H., Zhang, J., Walsh, J. E., Roesler, E. L., Hillman, B., Ballinger, T. J., and Weijer, W.: Arctic cyclones have become more intense and longer-lived over the past seven decades, Commun Earth Environ, 4, 348, https://doi.org/10.1038/s43247-023-01003-0, 2023.

---

## Author Comment (AC3)

wThe authors are grateful to the editor and all reviewers for their time and energy in providing helpful comments that have improved the manuscript. In our revised paper, we re-checked all revisions and performed grammatical corrections to help readers understand our manuscript easier.

In this document, reviewer' comments have been answered point by point. Referee comments are shown in black italics and author responses are shown in blue regular text and revised version of the manuscript is shown in green text.

**Reviewer #2:**

**General comments:**

The manuscript describes a great data set and a laudable effort to construct such data base of PL and MPC tracks based on ERA5 and satellite data. However, the characteristics and hence value of the data set is scientifically unclear. For existing similar track data sets, it is investigated how these are matched. It occurs that only a marginal set of points in the data base is characterized in the manuscript by these existing sets. Moreover, these appear the easiest tracks to capture, hence the value of most of the tracks remains unclear. This is associated with the fact that I miss a critical scientific assessment of the tracks generated. The manuscript appears subjective, rather than rigorous. There are ways to verify PL and MPC tracks with observations of atmospheric dynamics, in particular wind scatterometers. The use of scatterometers in this manuscript is rather unclear from a dynamic perspective and poor. In the least, the manuscript should be scientifically clarified and the pros and cons of the methodology better stipulated. In addition, a section on future work appears appropriate as much remains unclear in my interpretation of the manuscript.

Dear Prof. Stoffelen, we deeply appreciate your rigorous and constructive critique. Your insights have been instrumental in refining our methodology and dataset validation. Below, we address each point systematically.

**Major revision**

1. **Addition of hourly corrected ERA5 wind variables to the dataset**

To address the concern "The use of scatterometers... is unclear and poor", we adopted the recommended L4 product (WIND_GLO_PHY_L4_MY_012_006) to construct a new hourly 10-m vorticity dataset. Preliminary matching for January–April of 2001 indicates that ~90% of ERA5 vortices align with surface vorticity signatures. Full data processing is ongoing, as it will still take some time to download the data.. This approach will quantify ERA5's false alarm rate (e.g., Line 475)

Figure 1 exemplifies a matched case: the cyclone is embedded southeast of a synoptic-scale cold vortex moving southwestward from west of Svalbard to Greenland. The surface wind field, vorticity, and divergence distributions reveal a distinct frontal zone northwest of the cyclone, consistent with characteristics of typical PLs. Furthermore, the surface vorticity track aligns well with the 850 hPa vorticity track and exhibits greater smoothness, indicating that this dataset indeed provides finer dynamical features.

However, PL genesis mechanisms vary (Rasmussen & Turner 2003); surface vorticity validation may be insufficient for upper-level-triggered PLs (Blechschmidt et al. 2009; Yanase et al. 2016). Thus, we are integrating ERA5, scatterometer winds, AVHRR, and multi-source data to build a comprehensive validation framework.

[Figure]

Figure 1 (a) 10 m relative vorticity from WIND_GLO_PHY_L4_MY_012_006, (c) divergence, (d) wind field, and (b) 850 hPa relative vorticity from ERA5. Uniform smoothing was applied with radii of (a) 45 km and (b) 60 km. Orange (10 m) and blue (850 hPa) dotted lines denote matched vortex tracks, while black and green stars mark their center positions at 21:00 UTC on 24 February 2001.

**2. Clarification on AVHRR data availability**

Regarding the low proportion of vortices exhibiting cyclonic cloud features in AVHRR (also noted by Reviewer #3), we quantified AVHRR availability for vortex points and tracks. The results show that:

After excluding vortex tracks with >60% land presence (~20% reduction), 47,167 tracks remained for AVHRR matching. Matching required: (1) full 200-km radius coverage for individual points, (2) ≥2 matched points within ±3h of peak vorticity and ≥6 points per track lifetime. Figure 2 shows wintertime (Nov-Apr) matching statistics: 43% of points and 61% of tracks matched on average. Only ~3% of matched tracks were incorporated into the IMPMCT dataset. This low inclusion rate stems from cloud obstruction, cloud-ice contrast limitations, temporal resolution constraints, and detection methodology (e.g., higher 2001 inclusion reflects meticulous manual identification, while 2023's lower rate resulted from post-publication incidental discoveries). Crucially, IMPMCT's cyclone proportion underestimates true PMC prevalence, as many low-cloud even no-cloud PMCs lack discernible features.

While AVHRR covers relatively few cases, our dataset aims to provide a multi-source, high-accuracy collection—particularly those with clear cloud features—to aid users in understanding these phenomena (e.g., for model studies of PL-related clouds).

[Figure]

Figure 2: Annual winter (November-April) time series: (a) ERA5-derived vortex points (green), available AVHRR files (red), and AVHRR-matched vortex points (blue). (b) Ratio of AVHRR-matched vortex tracks to ERA5-derived tracks (yellow), and ratio of IMPMCT tracks to AVHRR-match tracks (purple). Note: Bars represent distinct categories (not stacked).

**3.  Explanation of mismatches with existing datasets**

We thoroughly compared IMPMCT with existing PL datasets and added specific analyses of mismatches with the Rojo list and Stoll's PL tracks:

To further investigate mismatches between the reanalysis-based tracks and existing PL datasets, we implemented a nearest-point matching analysis (Table 2). A successful nearest-point match was recorded when a PL center from any list had at least one co-temporal vortex center within 120 km (60 km for the Stoll dataset). The track-level mismatches primarily stemmed from these point-level discrepancies. Crucially, the methodological differences between datasets explain the variation: While the Noer list derives from numerically modeled and AVHRR-assimilated hourly positions (typical of operational forecasting systems), the Rojo list relies on direct AVHRR identification at irregular temporal intervals, resulting in greater deviation from ERA5 representations. Furthermore, the Rojo compilation includes numerous secondary PL centers—features inherently less resolved by reanalysis data (Stoll, 2022)— whereas Noer focuses primarily on dominant PLs of operational significance. This distinction is clearly reflected in our analysis: Major PL centers (n=2,527) exhibited an 80% matching rate, while secondary centers (n=1,115) showed significantly lower alignment (54%), thereby reducing Rojo's overall match rate.

For the Stoll dataset, we additionally calculated a vortex matching rate (Table 2), counting a match when a Stoll center fell within the spatial domain of its nearest co-temporal vortex. This metric primarily addresses positional offsets caused by vorticity peak misalignment, which appears attributable to differences in smoothing algorithms (illustrated in Fig. S2). Our implementation seems to employ stronger uniform smoothing compared to Stoll's methodology, explaining why more lenient identification thresholds yield superior track matching with Stoll's dataset. This provides a new insight for applying the algorithm. Although the algorithm is not

highly sensitive to the specific input vorticity fields, provided their grid spacing is sufficient to capture mesoscale vortices, the smoothing method also constitutes a significant factor contributing to variations in the identification results, alongside the identification parameters discussed in the sensitivity experiments (sect. 3.1.1). The smoothing approach should be specifically adapted to the assimilation noise and effective resolution of the input vorticity field. For instance, Gaussian smoothing may be preferable for model data with lower noise levels, as it more effectively preserves the positions of vortex cores.

[Figure]

Figure 1: (a) 850 hPa relative vorticity field obtained by ERA5 data. (b) AVHRR infrared imagery concurrent with the time step in (a). The shading represents 850 hPa relative vorticity smoothed over a uniform 60 km radius and local vorticity maxima are identified by green star symbols, while regions enclosed by solid black contours denote their borders. The red star symbol marks a mismatched cyclone center from Rojo's PLs list, while the black star symbol marks the nearest local vorticity maxima from the cyclone center (227 km).

Table 2: the matching rate of the reanalysis-based track dataset for IMPMCT generation compared to other PL track datasets.

| PL tracks | Time period | Tracks in Nordic Sea (>3hr) | Track matched fraction(%) | Points | Nearest points matched fraction(%) | Vortex matched fraction(%) |
|---|---|---|---|---|---|---|
| Noer | 2002-2011 | 114 | 87 | 1670 | 85 | - |
| Rojo | 2000-2019 | 370 | 69 | 3642 | 71 | - |
| Stoll | 2000-2020 | 3179 | 93.68 | 75650 | 93 | 99 |

[Figure]

Figure S2: ERA5 850-hPa fields: (a) Relative vorticity. (b) Uniform 60-km smoothed vorticity. Vorticity field comparison showing center displacement between Stoll (blue points) and our detection (green points).

**4. Comprehensive validation of the dataset**

Since all tracks follow identical generation procedures, unvalidated PMC tracks share consistency with verified ones. Nevertheless, validation remains essential. We added an overall track characterization:

For most newly identified mesoscale cyclones not present in other PL lists, a direct validation approach involves applying objectively derived PL identification thresholds from prior studies to independently verify three key characteristics: polar environment, mesoscale size, and cyclonic intensity:

1) Polar-front criterion: Since PMCs are defined as mesoscale cyclones forming north of the polar front (Rasmussen and Turner, 2003), we employ two indicators to distinguish polar air masses from extratropical air masses: Tropopause Potential Temperature ($\theta_{trop}$) and the Maximum poleward value of 200 hPa wind speed ($U_{200,p}$). For each cyclone, we compute the track-averaged $\theta_{trop}$ averaged within a 250 km radius of the cyclone center and the track-averaged $U_{200,p}$ within ±1.0° great-circle distance longitude. Stoll (2022) defined $\theta_{trop} < 300.8$ K as indicative of polar air mass origin for PLs, effectively distinguishing them from extratropical cyclones with a high retention rate (76%) across subjective archives while preserving 90% of known PLs. Han and Ullrich (2025) used $U_{200,p}$ (WIND200MAX) < 25 m s$^{-1}$ to position PLs north of the polar jet, achieving an ~80% hit rate for PL classification with a miss rate of only 11.9%.

2) Mesoscale-size criterion: Vortex radius calculated from the vorticity field is used to exclude extratropical cyclones penetrating polar regions and large-scale frontal structures. In Stoll (2022), a maximum vortex diameter of 430 km (representing the 90th percentile across all PL lists) was applied, excluding approximately 24% of non-PL vortices. As we employ the same vorticity boundary threshold ($1.0 \times 10^{-4}$ s$^{-1}$) for vortex definition, this criterion remains valid for our dataset.

3) Cyclonic intensity criterion: An effective metric for characterizing mesoscale cyclone intensity is the Pressure anomaly ($p_{def}$), defined as the difference between the mean Sea Level Pressure (SLP) within a 110 km radius and the SLP at the cyclone centre ($p_{def} = \overline{SLP}_{110km} - SLP$). Stoll (2018) demonstrated that high $p_{def}$ values (90% of PLs > 0.4 hPa) highlight the anomalous intensity of the local low-pressure centre relative to its environment, signifying a steep pressure gradient near the core, indicative of small, deep low-pressure systems typical of PLs. We calculate the maximum $p_{def}$ based on the SLP centre for each vortex track. For tracks where no SLP centre is identified, $p_{def}$ is set to 0.

All discriminatory features for IMPMCT tracks are computed from ERA5 data. The quantiles of these features and the proportion of tracks meeting each criterion are presented in Table 4. Notably, 88.4% of tracks satisfy the polar-front criterion, 90% meet the mesoscale criterion, and 84% fulfill the cyclonic intensity criterion. It is important to note that these thresholds were developed based on the more intense subset of PLs. For the broader spectrum of PMCs, the thresholds for $\theta_{trop}$ and $p_{def}$ are inherently stricter, as they correspond to the cold air outbreak environments and stronger destructive potential typically associated with PLs. Consequently, the vast majority of tracks in IMPMCT satisfy these validation criteria. Furthermore, the hourly time series of these discriminatory features are included in the dataset as auxiliary information to facilitate targeted case selection for user research.

Table 1: Quantiles of discriminatory features and proportion of IMPMCT tracks meeting validation criteria.

| criterion | Track feature | percentage | | | Proportion meeting the criterion (%) |
|---|---|---|---|---|---|
| | | 50% | 75% | 90% | |
| **Polar front** $\theta_{trop} < 301$ K or $U_{200,p} < 25$ m s$^{-1}$ | $\theta_{trop}$ (K) | 298.9 | 304.1 | 310.0 | 88.4 |
| | $U_{200,p}$ (m s$^{-1}$) | 18.4 | 23.7 | 29.7 | |
| **Mesoscale** $r < 215$ km | $r$ (km) | 137.1 | 176.9 | 213.5 | 90 |
| **Cyclonic** $p_{def} > 0.4$ hPa | $p_{def}$ (hPa) | 1.41 | 2.26 | 3.18 | 84 |

**Minor Revision**

1. Line 95: These images are not so clear. In a): Could a PMC also be in (8,74), (36,77) or (36,77)? Why not? In b): Could the PL also be in (34,76)? Why not?

**Re:** Thank you for highlighting this. We acknowledge that cyclone center identification involves a degree of subjectivity. The revised manuscript now explicitly states: The centers of comma cloud and spiral cloud configurations were determined visually following Forbes and Lottes (1985), based on the characteristic curvature and convergence of cloud bands surrounding the circulation core in satellite imagery.

Consistency with Rojo's centers is high (90% within 60 km; see Minor Revision #28).

2. Line 96: The ERA5 grid distance is 31 km, hence good dynamical representation will at most be 150 to 300 km following typical dynamical closure procedures. Is that good enough for PL/PMCs?

**Re:** We appreciate this clarification. The text has been revised to: With the improved resolution of reanalysis datasets, their ability to capture PLs has progressively advanced (Laffineur et al., 2014; Smirnova and Golubkin, 2017) …

3. Line 109: Belmonte Rivas and Stoffelen also suggest some other reasons for poor PL/PMC representation in ERA5: lack of transient variability, lack of divergence, lack of resolution; it appears of interest to mention these aspects.

**Re:** We have incorporated your insight to better describe ERA5's shortcomings: However, ERA5 significantly underestimates near-surface wind speeds within PL-affected regions (Gurvich et al., 2022; Haakenstad et al., 2021), attributed in part to insufficient representation of transient wind variability, surface divergence, and unresolved mesoscale features (Belmonte Rivas and Stoffelen, 2019). This limits its ability to objectively capture PLs' high-wind characteristics, thereby introducing notable limitations.

4. Line 113: Having looked at many collocated IR and scatterometer wind vector fields (e.g., here below), I have some problem with the terminology "cyclonic cloud feature". Cyclonic cloud features might occur due to closed surface circulation (cyclone definition) indeed, while wind shear conditions may also generate clouds in circles shapes on the mesoscales. Moreover, a cyclone may also exist in an abundance or lack of clouds in which a cyclone is not recognized in an IR image. In the image below (from today) circular cloud patterns are present on the left hand side, while the streamlines of the vector winds do not coincide with the cloud streaks. On the other hand, a cyclonic wind feature appears on the right side

of the plot, but where high clouds cover the wind structure below. This is today's example, while examples of apparent IR cloud mismatch with ocean vector winds occur almost every day on this site, in particular at high latitude.

**Re:** Thank you for your suggestion. The cyclonic clouds identified in this work are primarily based on: 1) typical PMC cloud morphologies described in Forbes and Lottes (1985) and PL cases from the Noer list; 2) corresponding ERA5 vortex tracks (>6 hr); 3) statistical validation showing >80% of cyclones exhibit strong surface lows ($p_{def}$ > 0.4 hPa; Table 4). Therefore, the cyclonic clouds in the dataset do possess cyclonic characteristics. We further note: Crucially, IMPMCT's cyclone proportion underestimates true PMC prevalence, as many low-cloud even no-cloud PMCs lack discernible features.

5. Line 170: remove "resolution"; Skamarock (2004) defines effective resolution as 5-10 times the grid distance of an atmospheric circulation model, due to the necessary dynamical closure for numerical stability of the model.

**Re:** The term "resolution" has been removed as suggested.

6. Line 172: Note that in particular the initiation of PMCs and PLs in ERA5 is brought by wind scatterometers as can be observed in time sequences at https://scatterometer.knmi.nl/tile_prod/index.php. Hence ERA5 PMCs/PLs may be biased to the availability of the satellite data used, which could be problematic in time series analyses of PMCs/PLs. As readers may not be generally aware of this dependency, it is better to state it.

**Re:** We emphasize this dependency in the Discussion: It is noteworthy that while this study demonstrates ERA5 reanalysis data's enhanced capability in capturing PMCs and PLs, it does not reflect ERA5's predictive skill for such systems. This predictive capability should be evaluated by testing ERA5 background states in characterizing PLs/PMCs, thereby isolating the influence of real-time assimilated data—particularly scatterometer measurements (Furevik et al., 2015).

7. Line 173: with a spatial resolution -> on a spatial grid

**Re:** Revised to: "on a spatial grid".

8. Line 182: To refer to scatterometer accuracy, one may use Vogelzang and Stoffelen (2022).

**Re:** Added scatterometer accuracy citation (Vogelzang and Stoffelen 2022): These advanced instruments are specifically engineered to deliver accurate(e.g., ASCAT-A zonal/meridional wind component error standard deviations of ~0.37/0.51 m s$^{-1}$ and ASCAT-B of ~0.39/0.44 m s$^{-1}$, Vogelzang and Stoffelen 2022), high-resolution, continuous wind vector measurements under all weather conditions, offering comprehensive global coverage of near-surface wind patterns.

9. Line 192: ASCAT-A, -B and -C have been operational since 2007.

**Re:** Corrected ASCAT operational timeline: QuikSCAT operated from 1999 to 2009, whereas ASCAT start operating since 2007.

**10.** Line 197: with stable spatiotemporal resolution -> exploiting the evolving global observing system; I.e., not necessarily of stable spatiotemporal resolution effectively, since depending on the initialization of small scales by observations, when available.

**Re:** Revised to "exploiting the evolving global observing system": To establish a more comprehensive cyclone track dataset in the Nordic Seas, we first utilize ERA5 reanalysis data exploiting the evolving global observing system to obtain all vortex tracks.

**11.** Line 208: Scatterometers measure the surface wind vector field and hence curl and divergence. See, e.g., Belmonte Rivas and Stoffelen (2019). King et al. (2022) found that tropical divergence as measured by scatterometers is closely related to moist convection. Similarly, one would expect that cyclonic disturbances are very well depicted in curl and divergence. These are furthermore available at https://data.marine.copernicus.eu/product/WIND_GLO_PHY_L4_MY_012_006/description. It also provides hourly corrected ERA5 wind variables for reference. Why not put them in the database? They provide a stable reference over time as each instrument product does not change over time.

**Re:** Thank you for your suggestion. As outlined in Major Revision #1 regarding the process of adding hourly corrected ERA5 wind variables to the dataset, these dynamical characteristics will be included in the dataset once the data download is complete.

**12.** Line 232: The vorticity field appears noisy as I understand the text. Nevertheless, no observations exist to initialize 4D dynamical structures well on scales below 100 km over the ocean, hence 60-km filtering may not be too problematic. The noise may be due to the fact that you use analyses, rather than more consistent dynamical model fields, i.e., background (first guess) ERA5 data as in Belmonte Rivas and Stoffelen (2019) for example. Reanalyses fields are affected by the observations being assimilated, using spatial structure functions, which are posed as stream function and velocity potential "blobs", defined based on forecast ensemble statistics. These increments may not treat vorticity fields well and produce noise. Another reason may be in interpolation of the vorticity fields, but where no details are provided.

**Re:** Thank you for your correction. It is necessary to introduce a potential source of vorticity perturbations in the text. The original text has been revised to: A uniform 60-km smoothing radius is applied to hourly 850-hPa relative vorticity to disconnect weak continuity zones and eliminate minor perturbation maxima, which may arise from assimilation increments (Belmonte Rivas and Stoffelen, 2019).

**13.** Line 314: All steps appear rather ad hoc, but together they define a vortex isolation and data procedure. Moreover, it appears as a community procedure, as others elaborated similar procedures. Does the procedure work similarly well for other reanalyses, mesoscale models or the operational ECMWF analysis? To me, it appears tuned to the characteristics of your input ERA5 fields. Perhaps mention that other meteorological model fields may require further tuning of the vortex detection procedure.

**Re:** Thank you for your comment. This algorithm is indeed a general procedure, with specific code written by ourselves and made public in the data and code availability section. In our response to Reviewer #1, we supplemented sensitivity experiments on the algorithm parameters (https://doi.org/10.5194/essd-2025-186-AC1). In addition to the above parameters, the selection of smoothing algorithms also has a certain impact on the identification results. In

Major Revision #3, we added a description of the algorithm's applicability: This provides a new insight for applying the algorithm. Although the algorithm is not highly sensitive to the specific input vorticity fields, provided their grid spacing is sufficient to capture mesoscale vortices, the smoothing method also constitutes a significant factor contributing to variations in the identification results, alongside the identification parameters discussed in the sensitivity experiments (sect. 3.1.1). The smoothing approach should be specifically adapted to the assimilation noise and effective resolution of the input vorticity field. For instance, Gaussian smoothing may be preferable for model data with lower noise levels, as it more effectively preserves the positions of vortex cores.

**14.** Line 316: established -> constructed
**Re:** Term replaced as suggested.

**15.** Line 318: Terrain-induced flows are normally tied to the terrain and not to the wind, hence presumably they'd typically not produce vortex tracks according to your criteria?
**Re:** Thank you for your suggestion. Terrain disturbances may still generate PLs (Kristjánsson et al., 2011), but our work does not separately classify such polar lows. The original expression may have been inaccurate, so we have revised it to: … including not only cyclonic systems but also low-pressure troughs, and small-scale atmospheric disturbance.

**16.** Line 320: established -> comparison; recall that AVHRR are not a direct measurement of PMC, cf. comment 113.
**Re:** Revised to: To assess whether these vortices represent PMCs, AVHRR infrared imagery is used for comparative validation.

**17.** Line 455: The concept of environmental wind speed bis not clear. What is its use? The 10m wind vector around a moving vortex is rather variable, depending on steering flow and vortex strengths. The baroclinic nature of these high-latitude vortices makes their surface appearance usually asymmetrical. I can understand you'd like to capture this, but this is not clear from the text. Please clarify what relevant dynamical characteristics can be extracted. Fig. 11b appears a vortex interacting with land and hence surface winds are distorted?
**Re:** Thank you for your suggestion. The calculation of environmental wind speed is primarily used to capture larger-scale atmospheric motions near cyclones, such as frontal zones and large-scale strong winds during cold air outbreaks. In the framework for constructing surface cyclonic circulation characteristics outlined in Major Revision #1, the concept of environmental wind speed will be replaced by surface cyclonic vorticity.

**18.** Line 458: To first order, the destructive force goes with the third power of the wind speed, irrespective of it is generated by the environmental flow, vortex contribution or related to local convection, all count. In open sea, the waves, build by the wind, are of course very important as well, as the dimensions of the structures at sea may resonate with long and forceful waves.
**Re:** Thank you for your suggestion. It is true that wind speed is positively correlated with destructive force. However, determining whether such strong winds are driven by the cyclone itself is also a key issue. Nevertheless, as mentioned in Major Revision #1, using surface vorticity will be a good indicator to describe the intensity of the cyclone itself.

19. Line 470: The scatterometer section is rather poor as scatterometers, in particular ASCAT, reveal detailed dynamical PL characteristics. Wind vectors fields reveal the exact surface position, structure and divergence and curl and with high coverage. See also comment 208. Unfortunately, not much has been published on active satellite surface winds and PLs, while Furevik et al. (2015) provide some overview.

Re: We acknowledge this gap and highlight future use of L4 winds (Major Revision #1).

20. Line 475: You find many tracks that are not in AVHRR. Following the comment above, this could well be because the vortical structure is not well expressed in the clouds. Observed dynamics at the surface may prove a better way to verify these vortices. A problem here is that scatterometer winds are only consulted after a imperfect AVHRR filter, rather than before this filter. This can be done by exploiting collocated model and scatterometer data and their spatial gradients, which are available. When only one scatterometer is available (up to 2007), then track cannot be well verified, but every occasion a vortex appears in a scatterometer swath verification may be done. That would results in hits and misses of ERA5 vortices, which verify your product more substantially in my view.

Re: Thank you very much for your suggestion. We clearly recognize that due to the spatiotemporal resolution of AVHRR and the challenge of polar remote sensing, AVHRR-based identification methods will miss a large number of PMC cases. However, for existing case studies, cloud images remain essential indicators of cyclone development and position. Therefore, to ensure the comprehensiveness of tracks, we still hope to retain PMC tracks verified by AVHRR comparison. However, ASCAT/QuikSCAT-based identification methods show great potential in our view. Therefore, as shown in the process outlined in Major Revision #1, the matching results between near-surface vortices and ERA5 vortices will be added to the revised manuscript, and this method will be mentioned in Future Work.

21. Line 476: "measurable wind patterns"? My experience in scatterometry for PL/MPC is that tracking is very well feasible and measurable. I copy below a slide I show in nowcasting training using https://scatterometer.knmi.nl/tile_prod/index.php . For a description, see the figure above. Several things to note here: 1) Many scatterometer acquisitions exist over a day to verify both model dynamics (green arrows) and IR images (grey-scale). 2) IR clouds follow the dynamics seen at the surface, i.e., the dynamics produce clouds in upward motion and dissolve clouds in downward motion, i.e., the clouds follow the winds. 3) Initially, a through appears in the scatterometer winds below a cloud shield, where the green arrows are not informed by it initially. As scatterometer winds are assimilated at ECMWF the disturbance appears in the model data over the day. As mentioned earlier, L3 and L4 products are produced with scatterometer information, model information, incl. ERA5, and fields of spatial derivatives. These appear more ideal to "measure" model and, after collocation, AVHRR characteristics in PL/MPC than the rather unfavorable diagnostics presented here.

Re: Thank you very much for your suggestion. Major Revision #1 shows a surface vortex track obtained using the tracking algorithm, which matches the 850hPa vorticity track. This is indeed highly feasible and measurable. Its effectiveness will be verified after processing all surface wind field data.

**22.** Line 486: 3 hours implies three points, right? 50% in these cases implies only 2 of 3 points and 80% 3 of 3 points and 4 of 5 hits for longer tracks for example. It is clear that adding more lenient vortex criteria will improve apparent skill as the Stoll data set is fixed. It does not necessarily imply better performance though as Stoll. How much false tracks/points do you add?

**Re:** Thank you for your comment. Table 2 shows the proportion of nearest matching points between vortex centers and other cyclones. For the PL lists of Stoll and Noer, the matching rate of vortex points in the reanalysis track dataset is sufficiently high because their datasets are assimilated by models and have hourly resolution.

However, for the Rojo list, which only uses AVHRR for identification, the temporal resolution of cyclone tracks can be several hours. Therefore, for the Rojo list, three points are often more than six hours apart, and it is more likely that vortices and cyclone centers are far apart, as shown in Figure 1. Hence, a 50% matching ratio threshold is appropriate. More importantly, the selection of a 50% matching ratio and a 150km separation threshold is intentionally set to be the same as in Stoll (2022) to facilitate testing the effectiveness of the vortex algorithm.

**23.** Line 493: demonstrates? Clearly, wind variability is high in cold air outbreaks near the surface and upper air interaction more fierce. Allowing more noise in ERA5 vorticity or more lenient vortex criteria will reveal more tracks, but are they reliable? If some of them appear in the proximity of observed tracks, it appears insufficient to demonstrate capability. How many unverified tracks are produced (false alarms)? Could these accidentally be added to the hit
list? In that case skill is not enhanced, but rather PL/MPC noise is added.

**Re:** Thank you for your comment. The following aspects demonstrate that the increase in matching rate is not primarily due to increased matching of cyclones by transient disturbances:

1) As shown in Major Revision #3, our algorithm exhibits stronger uniform smoothing than Stoll, which suppresses noise to a considerable extent.
2) Only vortex tracks (>6hr) exceeding 60% of the reference PL trajectory's lifespan are included in the matching process. We apologize for not elaborating on the third point in the text. In the revised manuscript, this condition has been added: Furthermore, to avoid incidental matches of transient spurious tracks to PLs, only vortex tracks with lifespans exceeding 60% of the reference PL trajectory's duration are included in the matching process.

**24.** Line 495: So, ERA5 finds about 10 times more PLs/MPCs than the most extensive observational data set (Rojo). Is this noise? Looking at your AVHRR score, noise appears indeed manifest; 57,688 ERA5 vortex tracks, only 1,184 or 1 in 50 are confirmed. This may be related to the fact that AVHRR is a rather indirect measure of vortical activity, while you appear to appreciate the skills of AVHRR. What are the >90% misses in your data set? As these amounts appear rather overwhelming, it appears very relevant to understand their characteristics if these are used for geophysical analyses or trend analyses. The difference with Stoll's 3179 tracks from the same ERA5 is also rather overwhelming. What are the differences? I further understand less than 700 (only about 1%) remain for further comparison. I'm concerned what the other 99% represent?

**Re:** Thank you for your comment. In Major Revision #2, we show that the proportion of vortex tracks verifiable by AVHRR is approximately 61%, and due to the image resolution issues of AVHRR, the proportion of truly verifiable vortices is likely even lower. For tracks not included in the IMPMCT track set, in addition to the lack of AVHRR track verification, there are several possible reasons:

1) Extratropical types, which appear too uniformly bright or large in cloud images and do not conform to the common cloud appearance of PLs/PMCs, thus being excluded;

2) No corresponding cyclonic cloud appearance or insufficiently obvious cyclonic cloud features, which occurs in cases of cloud obstruction,or cloud-free cyclones formed due to low water vapor;

3) High separation from ERA5 vortex tracks, with only cyclone tracks that maintain good consistency with the movement of vortex tracks being retained.

The above three situations indicate that the tracks not selected by the dataset do not represent false alarms; in other words, we only strictly retain samples with clear and unambiguous cloud images, and exclude those that are unclear or cannot be fully confirmed.

Additionally, regarding the false alarm rate of the ERA5 vortex track set, we plan to verify it using wind scatterometer data as described in Major Revision #1, which will be the focus of our next manuscript revision.

**25.** Line 507: demonstrate that such discrepancies are not errors -> characterize such discrepancies; they are errors as ERA5 uniquely represents PLs/MPCs.

**Re:** Revised to: "characterize these discrepancies". In Major Revision #3, we calculated the vortex matching rate with Stoll (99%), demonstrating that such mismatches are more likely caused by peak misalignment due to different smoothing algorithms.

**26.** Line 509: stable -> negligible

**Re:** Term replaced as suggested.

**27.** Line 512: remove "stable"; the choice of this word is a bit concerning, does it imply that you favor a smooth representation of disturbances? Spatial smoothing is applied, but it can obviously kill PLs/MPCs, which is a negative effect. If Stoll uses data from ERA5 that is less interpolated, it may in fact be a good thing that it represents more variability? Please elaborate in your manuscript.

**Re:** Thank you for your comment. It is indeed arbitrary to assume that a lower standard deviation indicates greater correctness. The word "stable" may imply a preference for smoothness, while greater variability may also be reasonable. In the revised manuscript, this description has been replaced with: Additionally, vortex property differences increase with distance, indicating that discrepancies between IMPMCT and Stoll tracks stem from differing identification thresholds. To further demonstrate that such discrepancies are not errors but reflect slightly different tracking paradigms, we calculate the standard deviation of vortex properties across three consecutive time steps for each track and computed track-wide averages. Consistently low-amplitude variations in vortex properties along a track suggest coherent feature identification by the respective method. Fig. 13 (b), (c) and (d) show the track-averaged local standard deviations of the three vortex properties for IMPMCT and Stoll datasets. Crucially, the magnitudes of these local variabilities in IMPMCT tracks are generally comparable to those in Stoll's tracks. This alignment in variability indicates that the increasing

property differences observed at larger separation distances arise from small-scale peak misalignments inherent to each method's detection logic, rather than representing erroneous identification by either approach. In fact, the IMPMCT variabilities are often slightly smoother, consistent with its specific algorithmic choices.

**28.** Line 532: Please indicate in the figure legend what percentage of the most favorable (matched) cases it represent. The non-matched cases are less detectable and probably have much less favorable verification.

**Re:** Thank you for your suggestion. Although we allow a 120 km separation, in fact, 90% of cyclones matched with Rojo are within 60 km, and this information has been added to the manuscript. For unmatched cyclones, in addition to cases where AVHRR data is unavailable, situations where ERA5 fails to capture them as shown in Major Revision #3 may also be reasons.

A total of 1432 cyclone centers from the Rojo list (corresponding to 140 cyclone tracks) were matched to the IMPMCT cyclone tracks. Notably, although the maximum allowable matching distance was set to 120 km, the 90th percentile of the matching distances was 55.97 km. This suggests minimal cyclone center identification errors when scan times could not be strictly aligned.

**29.** Line 541: "Despite" or "Due to"? Less favorable cases may not match well?

**Re:** Thank you for your suggestion. Owing to the inclusion of secondary PL centers (30% in Rojo; Major Revision #3), which exhibit lower detectability (54% match rate), overall consistency decreases.

**30.** Line 544: "reasonable"; you allow a 120 km separation and then one gets separations with a SDD of about 120 km, which implies little skill. Do you reason for little skill? Presumably, further work is needed to explain the lack of skill? Better explain to the users what further work would be appropriate in this discussion.

**Re:** Under the condition that the identification error of the center is low, such differences in cyclone scale are more likely due to inconsistent methods for measuring cyclone diameter. In the original text, we explain these differences: When identification errors of the cyclone center are low, the differences in diameter compared to the Rojo list stem not only from inconsistent measurement methods but also significantly from subjectivity. Due to the frequent presence of frontal cloud bands associated with cyclones, consistent measurement of the cyclone's long axis proves challenging. Furthermore, when a cyclone is adjacent to other cloud systems, its extent is often subject to ambiguity.

In addition, these characteristics are provided as reference quantities, and users can adjust them according to the given cloud image examples.

[Figure]

Figure 2: Frequency distribution of bias in (a) Track-max near-surface wind speed and (b) diameter between matched cyclones in the Rojo and IMPMCT datasets (Rojo minus IMPMCT). The cyclone diameter in IMPMCT is calculated as the average of the width and length of the bounding box enclosing the cyclone.

**31.** Line 559: How do you know what these cases are? They have not been verified, at least not in the manuscript. Could they not be numerical artefacts? Are they associated with real features or are these ERA5 simulated features?

Re: Thank you for your comment. Our tracks do include longer segments relative to Stoll's track set, which is due to more lenient thresholds. However, in our individual observations, we confirm that these segments are sufficiently closely connected; otherwise, there would definitely be abrupt changes in vortex properties, which would be reflected in the local standard deviation mentioned in Minor Revision #27. All tracks are matched with cyclone images. For track segments beyond the coverage of cyclone images, there seems to be no good ways to confirm their authenticity. After obtaining the matching of near-surface vorticity as shown in the process in Major Revision #1, these segments may be better verified.

**32.** Line 565: Is the point not how reliable ERA5 is to represent PLs and PMCs? One could test that using the cases where verification is available and determine and not yet used in ERA5 (by data assimilation). Therefore, testing ERA5 background states, winds are independent of any new observations, one could establish the capability of ERA5 to predict PLs and MPCs. Only after this, ERA5 can be used with confidence for associated geophysical studies in my view. Would you agree?

Re: Thank you for your suggestion. We agree that testing ERA5's ability to capture PLs and PMCs using background fields rather than assimilated fields can verify ERA5's predictive capability. As shown in Minor Revision #6, we emphasize this fact in the discussion section.

**33.** Line 580: As explained above, observations directly associated with PL/MPC dynamics may be further exploited to characterize these systems and the fidelity of reanalyses to represent them.

Re: Thank you for your suggestion. As shown in Major Revision #1, dynamical characteristics will be added to the IMPMCT dataset soon as an important supplement to surface circulation characteristics.

**Reference**

Blechschmidt, A. -M., Bakan, S., and Graßl, H.: Large-scale atmospheric circulation patterns during polar low events over the nordic seas, J. Geophys. Res., 114, 2008JD010865, https://doi.org/10.1029/2008JD010865, 2009.

Yanase, W., Niino, H., Watanabe, S. I., Hodges, K., Zahn, M., Spengler, T., and Gurvich, I. A.: Climatology of polar lows over the sea of japan using the JRA-55 reanalysis, Journal of Climate, 29, 419–437, https://doi.org/10.1175/JCLI-D-15-0291.1, 2016.

Belmonte Rivas, M. and Stoffelen, A.: Characterizing ERA-interim and ERA5 surface wind biases using ASCAT, Ocean Sci., 15, 831–852, https://doi.org/10.5194/os-15-831-2019, 2019.

Furevik, B. R., Schyberg, H., Noer, G., Tveter, F., and Röhrs, J.: ASAR and ASCAT in polar low situations, Journal of Atmospheric and Oceanic Technology, 32, 783–792, https://doi.org/10.1175/JTECH-D-14-00154.1, 2015.

Forbes, G. S. and Lottes, W. D.: Classification of mesoscale vortices in polar airstreams and the influence of the large-scale environment on their evolutions, Tellus A, 37A, 132–155, https://doi.org/10.1111/j.1600-0870.1985.tb00276.x, 1985.

Kristjánsson, J. E., Thorsteinsson, S., Kolstad, E. W., and Blechschmidt, A.: Orographic influence of east greenland on a polar low over the denmark strait, Quart J Royal Meteoro

Soc, 137, 1773–1789, https://doi.org/10.1002/qj.831, 2011.

Stoll, P. J., Graversen, R. G., Noer, G., and Hodges, K.: An objective global climatology of polar lows based on reanalysis data, Quart J Royal Meteoro Soc, 144, 2099–2117, https://doi.org/10.1002/qj.3309, 2018.

Stoll, P. J., Spengler, T., Terpstra, A., and Graversen, R. G.: Polar lows – moist-baroclinic cyclones developing in four different vertical wind shear environments, Polar lows, 2021.

---

## Author Comment (AC4)

**Report on Use of Recommended L4 Product**

**Dear Prof. Stoffelen**

We are pleased to report on our use of the L4 product (WIND_GLO_PHY_L4_MY_012_006) you recommended. We successfully downloaded the hourly 10m wind field products, including vorticity and wind component variables, for November to April from 2000 to 2006. This data (hereafter referred to as CMEMS) was used to match all ERA5 and IMPMCT track points for the validation of surface cyclonic circulations. The methods for identifying surface vorticity centers and constructing tracks were identical to those used for the 850 hPa vorticity tracks described in the main text. To ensure the robustness of the matching results, three different parameter combinations were tested, and the corresponding matching results are presented in Table 1.

1) Experiment a: $R_{smth}$(uniform smoothing radius) = 60 km, $\zeta_{max0}$ = 0.8×10⁻⁴ s⁻¹, $\zeta_{min0}$ = 0.5×10⁻⁴ s⁻¹, $\gamma$ = 0.15

2) Experiment b: $R_{smth}$ = 45 km, $\zeta_{max0}$ = 0.8×10⁻⁴ s⁻¹, $\zeta_{min0}$ = 0.5×10⁻⁴ s⁻¹, $\gamma$ = 0.15

3) Experiment c: $R_{smth}$= 45 km, $\zeta_{max0}$ = 1×10⁻⁴ s⁻¹, $\zeta_{min0}$ = 0.5×10⁻⁴ s⁻¹, $\gamma$ = 0.25

Matching between 850 hPa and surface vortices was determined based on their actual geographical distance. For vorticity centers at the same time step, a match was identified if the distance was less than 100 km. For a track pair, the tracks were considered matched if the proportion of matched points exceeded 80% of the overlapping time period of both tracks, provided the overlap duration exceeded half the lifetime of at least one of the tracks.

**Table 1: Matching results between 850 hPa vorticity tracks and surface features for different parameter combinations.**

| Experiment | Points | Tracks (>3hr) | Points matched fraction(%) | | Tracks matched fraction(%) | | Mean wind speed bias (m/s) |
|---|---|---|---|---|---|---|---|
| | | | ERA5 | IMPMCT | ERA5 | IMPMCT | QuickScat-CMEMS |
| a | 378627 | 22750 | 46.1 | 77.4 | 33.0 | 88.0 | 5.2 |
| b | 520103 | 31589 | 51.6 | 79.8 | 36.6 | 88.5 | 5.4 |
| c | 388577 | 23652 | 43.1 | 74.0 | 32.1 | 86.0 | 5.4 |

As shown in Table 1, among the 12,030 ERA5 tracks and 200 IMPMCT tracks from 2000–2006, up to 88.5% of IMPMCT tracks and 79.8% of IMPMCT points were matched with surface vorticity. This proportion significantly exceeds the match rate for all ERA5 tracks during this period (36.6%). These results demonstrate that surface vorticity serves as an effective method for tracking and validating Polar Mesoscale Cyclones (PMCs). In other words, the dataset we previously provided, which primarily references cloud charts, has been demonstrated to be representative.

However, if the maximum wind speed within the surface vortex area is taken as the core

wind speed of the PMC points, the results show a significant low bias (approximately ~5 m/s, as shown in Table 1) compared to near-real-time wind speeds measured by QuikScat (n ≈ 200). Below, we present five matched cases (Figure 1-5) showing surface vorticity and the corresponding 850 hPa vorticity tracks from IMPMCT, along with comparisons of the associated ERA5 10m wind fields, CMEMS bias-corrected ERA5 10m wind fields, and QuikScat 10m wind fields. We find that compared to the near-real-time wind fields, the hourly averaged wind fields do not always adequately represent the explosively strong surface winds associated with the cyclones. While it is difficult to definitively conclude whether this discrepancy arises from an overestimation by QuikScat or an underestimation by the CMEMS corrected product, the wind difference fields (e.g. Figs. d of 1-5) often appear to align more closely with cloud features. This indicates a better correspondence between real-time wind fields and cloud imagery.

Although the WIND_GLO_PHY_L4_MY_012_006 product provides consistent, hourly, and bias-corrected ERA5 wind data that would greatly enhance the wind speed information within the IMPMCT dataset, integrating these corrected winds would require additional research to fully interpret the associated discrepancies. Moreover, incorporating derived dynamic features such as vorticity and divergence would introduce further complexity due to the inherent challenges in validating such parameters.

We sincerely acknowledge the value of this product and have carefully considered its inclusion. However, after thorough evaluation, we concluded that it would be more appropriate to address these challenges in a separate, dedicated study rather than incorporating the product into the current dataset. We highlight in the discussion section of our paper both the limitations in wind data resolution and the potential of this product for future applications such as cyclone validation. We believe it holds particular promise for supporting dynamic investigations in subsequent research:

"The dataset does not explicitly distinguish between PMCs and PLs due to the time-sparse wind speed data, particularly when the cyclone's wind speed at a given time step falls below the 15 m s$^{-1}$ threshold. In such cases, it is difficult to determine whether the cyclone is a PMC or merely in a weaker phase of a PL. **A more reliable validation method may be provided by the hourly bias-corrected sea surface wind product from the E.U. Copernicus Marine Service Information (https://doi.org/10.48670/moi-00185). This product systematically corrects ECMWF ERA5 model fields using scatterometer observations to reduce persistent biases and includes uncertainty estimates.**"

Three CMEMS track-datasets are stored at: https://github.com/thebluewind/IMPMCT. We sincerely appreciate your valuable comments and suggestions, particularly the recommendation of this product. It will be the dataset of choice for our subsequent statistical work investigating the development mechanisms of PMCs and PLs.

Sincerely,

[Figure]

**Figure 1: Vorticity matching results. (a) 10m vorticity and wind fields from the WIND_GLO_PHY_L4_MY_012_006 product. (b) 850 hPa vorticity and 10m wind fields from ERA5 hourly data. (c) Brightness temperature image from AVHRR channel 4 and 10m wind field data from QuickSCAT. (d) Wind speed difference: (c) minus (a). Black and red stars represent matched 10m and 850 hPa vorticity points, respectively. Orange and blue dotted lines represent matched 10m and 850 hPa vorticity tracks. The legend in (c) shows the AVHRR and QuickSCAT scan times, as well as the cyclone (vortex) core maximum wind speeds retrieved from QuickSCAT and CMEMS wind fields.**

[Figure]

**Figure 2: Same as Figure 1**

[Figure]

**Figure 3 : Same as Figure 1**

[Figure]

**Figure 4 : Same as Figure 1**

[Figure]

**Figure 5 : Same as Figure 1**

---

## Author Response (AR2)

The authors are grateful to the editor and all reviewers for their time and energy in providing helpful comments that have improved the manuscript. In our revised paper, we re-checked all revisions and performed grammatical corrections to help readers understand our manuscript easier. In this document, reviewer' comments have been answered point by point. Referee comments are shown in black italics and author responses are shown in blue regular text and revised version of the manuscript is shown in green text.

**General comments:**

My assessment is that the manuscript has been substantially clarified and only have one remaining point that requires further consideration. The data base contains scatterometer wind speeds, while for vorticity analysis the scatterometer spatial derivatives are of more interest as they are closely associated with deep moist convection and cyclonic activity. It would in fact make a lot of sense to use these observed fields for a PL/PMC identification, rather than ERA5, as ERA5 is rather coarse (Belmonte Rivas et al., 2019). The instantaneous curl and divergence data are available as CMEMS L3 data (while CMEMS L4 contains corrected ERA5 winds).

It would hence be of great interest to add these fields to the data base, while I understand that this is a lot of work (resulting in a new manuscript I guess). Nevertheless, some of the paragraphs on scatterometry should be adapted to embrace this potential, while in the current version rather the regretful limited use of the full potential appears to be highlighted.

This manuscript stresses the use of scatterometer wind speed (not vector or spatial derivatives) and SLP. SLP is obviously associated with the scatterometer wind vector field, while mainly depicting the barotropic component. Scatterometer vector fields and spatial derivative observations are hence more meaningful to depict PLs/PMCs than SLP or other model-based variables.

**Re:** We sincerely appreciate your recognition of the revised manuscript and the time and effort you dedicated. Furthermore, your reminder made us realize the importance of scatterometers, particularly their derivatives, for characterizing the dynamic features of PMCs/PLs, as well as the concerns regarding ERA5's ability to consistently and systematically represent PMCs. These points will be further emphasized in the text.

**Minor revision**

198: Vogelzang and Stoffelen (2022) appears in the text, but not in the references.

**Re:** Thanks. We have added this literature to the reference list.

200: wind patterns; here divergence and vorticity (curl) patterns can be added as these are produced as L3 CMEMS products. Moreover, I noted earlier that King et al. (2022) show their intimate relationship with deep convection processes, albeit in the tropics. It of interest to highlight this.

**Re:** Thank you for your suggestion. We have emphasized here that vorticity and divergence are equally important information alongside direct wind speed data: These advanced instruments are specifically engineered to deliver accurate (e.g., ASCAT-A zonal/meridional wind component error standard deviations of ~0.37/0.51 m s-1 and ASCAT-B of ~0.39/0.44 m s-1, Vogelzang and Stoffelen 2022), high-resolution, continuous wind vector measurements

under all weather conditions, offering comprehensive global coverage of near-surface wind patterns. The full potential of these measurements extends to their spatial derivatives, specifically vorticity and divergence, which are closely associated with deep moist convection and cyclonic activity (King et al., 2022).

488: "what is retrieved from scatterometer wind measurements may not always reflect cyclone-induced circulation, but could also include contributions from large-scale advective wind". As curl is well measured by scatterometers as well as divergence, recognizing vortices in scatterometers in a strong background flow is not difficult at all. I suggest: "scatterometer wind speeds may not always reflect cyclone-induced circulation and could include contributions from large-scale advective wind. By using the spatial derivatives from scatterometer wind vector fields, vortical structures or divergent flows near the surface associated with PLs/PMCs become easily visible (King et al., 2022)".

**Re:** Thank you for your suggestion. We have corrected our statement in the original text to describe the importance of vorticity and divergence: It is important to recognize that scatterometer wind speeds may not always reflect cyclone-induced circulation and could include contributions from large-scale advective wind. By using the spatial derivatives from scatterometer wind vector fields, vortical structures or divergent flows near the surface associated with PLs/PMCs may become easily visible (King et al., 2022).

498-511: Fig.12 remains unconvincing in my view, as divergence and curl from the scatterometer are not explicitly shown. They are available in the CMEMS wind TAC: https://doi.org/10.48670/moi-00183. As probably exists in ERA5 for this case, I imagine also the surface winds show an elongated vortical structure and certainly convergence associated with the weak disturbance in Fig. 12b. It could also be that ERA5 is a bit too enthusiastic with vortical signal and coincidently AVHRR shows remnants of a deceased disturbance? From my experience, the link between model winds and IR imagery is weaker than the link between surface vector winds and imagery. Please reconsider the text and the figure or remove it altogether.

**Re:** Thank you for your suggestion. For Fig.12, we have calculated and supplemented the more direct vorticity distribution, and emphasized the value of the spatial derivatives of scatterometer data. The case shown in Fig.12b is indeed the early stage of a cyclonic process, perhaps a moment when clouds appear before the surface circulation becomes fully established; therefore, we have also added a brief supplementary explanation:

It is important to recognize that scatterometer wind speeds may not always reflect cyclone-induced circulation and could include contributions from large-scale advective wind. By using the spatial derivatives from scatterometer wind vector fields, vortical structures or divergent flows near the surface associated with PLs/PMCs may become easily visible (King et al., 2022). For instance, Fig. 12a illustrates a system with a well-defined cyclonic circulation where the high wind speeds at its head are clearly associated with the cyclone itself. The fine-scale and complex structure of the corresponding vorticity field exhibit a strong and organized vorticity signature coincident with the cloud vortex, confirming the presence of an intense mesoscale vortex and a trailing shear line. In contrast, Fig. 12b shows a

case where the wind field is largely straight and convergent in the ambient flow, accompanied by only a weak vorticity signal (1×10-4 s-1) localized near the cloud eye and lacking any broader organized cyclonic structure, suggesting that the surface circulation appears to be either not yet formed or obscured. Due to technical constraints, additional parameters such as vorticity and divergence are not provided alongside wind speed. Nevertheless, they retain substantial application potential, as evidenced by the vorticity structures revealed in Fig. 12, which demonstrate the value of scatterometer spatial derivatives in elucidating the complex dynamical features of mesoscale systems.

Figure 12: VCI images overlaid with near-surface wind speeds for cyclones exhibiting strong (a) and weak (b) cyclonic near-surface wind patterns. Color shading represents QuickSCAT-measured 10m near-surface wind speeds, with green arrows indicating corresponding wind vectors. Yellow borders denote the cyclones' bounding oriented box. Blue circular border represents the search range. Yellow and red stars indicate the cyclone center and maximum wind speed point locations. The vorticity calculated from the wind fields is shown as white-to-red contours, with units of  $10^{-4}$  s-1.

730: Here I suggest to promote the L3 scatterometer products containing the spatial derivatives of the instantaneous scatterometer wind vector observation fields. Certainly, the current golden age of wind scatterometry would help the future data base, if maintained.

Re: Thank you for your suggestion. In the discussion section describing the CMEMS wind products, we have emphasized the unique value of scatterometer spatial derivatives: In such cases, a more reliable validation method may be provided by the hourly bias-corrected sea surface wind product from the E.U. Copernicus Marine Service Information (CMEMS, <a href="https://doi.org/10.48670/moi-00185">https://doi.org/10.48670/moi-00185</a>). Such product systematically corrects ECMWF ERA5 model fields using scatterometer observations to reduce persistent biases and includes uncertainty estimates. Furthermore, the L3 scatterometer products available through CMEMS, which contain the spatial derivatives of the wind vector fields (vorticity and divergence), offer a more direct characterization of the dynamical core of mesoscale systems. These observed fields hold significant potential for refining objective identification criteria, moving beyond a reliance on wind speed thresholds alone.

765: I sincerely doubt that this resource permits establishment of comprehensive objective identification criteria based on reanalysis data, thereby enabling robust analysis of climate-scale trends and genesis potential shifts in PL/PMC activity. We've noticed that ERA5 PLs/PMCs are strongly affected by the presence of scatterometers for data assimilation and, similarly, other observing systems may affect mesoscale activity trends through variable sampling over the ERA5 reanalysis period. Please remove this statement that is not proven in this manuscript.

**Re:** Thank you for your suggestion. We fully understand your concern. The original text has now been removed.

**Reference:**

King, G. P., Portabella, M., Lin, W., and Stoffelen, A.: Correlating extremes in wind divergence with extremes in rain over the tropical atlantic, Remote Sensing, 14, 1147, https://doi.org/10.3390/rs14051147, 2022.